# On the Selection of Initialization and Activation Function for Deep Neural Networks

## Abstract

The weight initialization and the activation function of deep neural networks have a crucial impact on the performance of the training procedure. An inappropriate selection can lead to the loss of information of the input during forward propagation and the exponential vanishing/exploding of gradients during back-propagation. Understanding the theoretical properties of untrained random networks is key to identifying which deep networks may be trained successfully as recently demonstrated by Schoenholz et al. (2017) who showed that for deep feedforward neural networks only a specific choice of hyperparameters known as the 'edge of chaos' can lead to good performance. We complete this analysis by providing quantitative results showing that, for a class of ReLU-like activation functions, the information propagates indeed deeper for an initialization at the edge of chaos. By further extending this analysis, we identify a class of activation functions that improve the information propagation over ReLU-like functions. This class includes the Swish activation, $\phi_{swish}(x) = x \cdot \text{sigmoid}(x)$, used in Hendrycks & Gimpel (2016), Elfwing et al. (2017) and Ramachandran et al. (2017). This provides a theoretical grounding for the excellent empirical performance of $\phi_{swish}$ observed in these contributions. We complement those previous results by illustrating the benefit of using a random initialization on the edge of chaos in this context.

## 1 Introduction

Deep neural networks have become extremely popular as they achieve state-of-the-art performance on a variety of important applications including language processing and computer vision; see, e.g., LeCun et al. (1998). The success of these models has motivated the use of increasingly deep networks and stimulated a large body of work to understand their theoretical properties. It is impossible to provide here a comprehensive summary of the large number of contributions within this field. To cite a few results relevant to our contributions, Montufar et al. (2014) have shown that neural networks have exponential expressive power with respect to the depth while Poole et al. (2016) obtained similar results using a topological measure of expressiveness.

We follow here the approach of Poole et al. (2016) and Schoenholz et al. (2017) by investigating the behaviour of random networks in the infinite-width and finite-variance i.i.d. weights context where they can be approximated by a Gaussian process as established by Matthews et al. (2018) and Lee et al. (2018).

In this paper, our contribution is two-fold. Firstly, we provide an analysis complementing the results of Poole et al. (2016) and Schoenholz et al. (2017) and show that initializing a network with a specific choice of hyperparameters known as the 'edge of chaos' is linked to a deeper propagation of the information through the network. In particular, we establish that for a class of ReLU-like activation functions, the exponential depth scale introduced in Schoenholz et al. (2017) is replaced by a polynomial depth scale. This implies that the information can propagate deeper when the network is initialized on the edge of chaos. Secondly, we outline the limitations of ReLU-like activation functions by showing that, even on the edge of chaos, the limiting Gaussian Process admits a degenerate kernel as the number of layers goes to infinity. Our main result (4) gives sufficient conditions for activation functions to allow a good 'information flow' through the network (Proposition 4) (in addition to being non-polynomial and not suffering from the exploding/vanishing gradient problem). These conditions are satisfied by the Swish activation $\phi_{swish}(x) = x \cdot \text{sigmoid}(x)$ used in Hendrycks & Gimpel (2016),

Elfwing et al. (2017) and Ramachandran et al. (2017). In recent work, Ramachandran et al. (2017) used automated search techniques to identify new activation functions and found experimentally that functions of the form $\phi(x) = x \cdot \text{sigmoid}(\beta x)$ appear to perform indeed better than many alternative functions, including ReLU. Our paper provides a theoretical grounding for these results. We also complement previous empirical results by illustrating the benefits of an initialization on the edge of chaos in this context. All proofs are given in the Supplementary Material.

## 2 ON GAUSSIAN PROCESS APPROXIMATIONS OF NEURAL NETWORKS AND THEIR STABILITY

### 2.1 SETUP AND NOTATIONS

We use similar notations to those of Poole et al. (2016) and Lee et al. (2018). Consider a fully connected random neural network of depth $L$, widths $(N_l)_{1 \leq l \leq L}$, weights $W_{ij}^l \overset{iid}{\sim} \mathcal{N}(0, \frac{\sigma_w^2}{N_{l-1}})$ and bias $B_i^l \overset{iid}{\sim} \mathcal{N}(0, \sigma_b^2)$, where $\mathcal{N}(\mu, \sigma^2)$ denotes the normal distribution of mean $\mu$ and variance $\sigma^2$. For some input $a \in \mathbb{R}^d$, the propagation of this input through the network is given for an activation function $\phi : \mathbb{R} \to \mathbb{R}$ by

$$y_i^1(a) = \sum_{j=1}^{d} W_{ij}^1 a_j + B_i^1, \quad y_i^l(a) = \sum_{j=1}^{N_{l-1}} W_{ij}^l \phi(y_j^{l-1}(a)) + B_i^l, \quad \text{for } l \geq 2. \tag{1}$$

Throughout the paper we assume that for all $l$ the processes $y_i^l(.)$ are independent (across $i$) centred Gaussian processes with covariance kernels $\kappa^l$ and write accordingly $y_i^l \overset{ind}{\sim} \mathcal{GP}(0, \kappa^l)$. This is an idealized version of the true processes corresponding to choosing $N_{l-1} = +\infty$ (which implies, using Central Limit Theorem, that $y_i^l(a)$ is a Gaussian variable for any input $a$). The approximation of $y_i^l(.)$ by a Gaussian process was first proposed by Neal (1995) in the single layer case and has been recently extended to the multiple layer case by Lee et al. (2018) and Matthews et al. (2018). We recall here the expressions of the limiting Gaussian process kernels. For any input $a \in \mathbb{R}^d$, $\mathbb{E}[y_i^l(a)] = 0$ so that for any inputs $a, b \in \mathbb{R}^d$

$$\kappa^l(a, b) = \mathbb{E}[y_i^l(a) y_i^l(b)] = \sigma_b^2 + \sigma_w^2 \mathbb{E}_{y_i^{l-1} \sim GP(0, \kappa^{l-1})}[\phi(y_i^{l-1}(a)) \phi(y_i^{l-1}(b))]$$
$$= \sigma_b^2 + \sigma_w^2 F_\phi(\kappa^{l-1}(a, a), \kappa^{l-1}(a, b), \kappa^{l-1}(b, b)),$$

where $F_\phi$ is a function that depends only on $\phi$. This gives a recursion to calculate the kernel $\kappa^l$; see, e.g., Lee et al. (2018) for more details. We can also express the kernel $\kappa^l$ in terms of the correlation $c_{ab}^l$ in the $l^{th}$ layer used in the rest of this paper

$$q_{ab}^l := \kappa^l(a, b) = \mathbb{E}[y_i^l(a) y_i^l(b)] = \sigma_b^2 + \sigma_w^2 \mathbb{E}[\phi(\sqrt{q_a^{l-1}} Z_1) \phi(\sqrt{q_b^{l-1}}(c_{ab}^{l-1} Z_1 + \sqrt{1 - (c_{ab}^{l-1})^2} Z_2))]$$

where $q_a^{l-1} := q_{aa}^{l-1}$, resp. $c_{ab}^{l-1} := q_{ab}^{l-1}/\sqrt{q_a^{l-1} q_b^{l-1}}$, is the variance, resp. correlation, in the $(l-1)^{th}$ layer and $Z_1, Z_2$ are independent standard Gaussian random variables. when it propagates through the network. $q_a^l$ is updated through the layers by the recursive formula $q_a^l = F(q_a^{l-1})$, where $F$ is the 'variance function' given by

$$F(x) = \sigma_b^2 + \sigma_w^2 \mathbb{E}[\phi(\sqrt{x} Z)^2], \quad \text{where} \quad Z \sim \mathcal{N}(0, 1). \tag{2}$$

Throughout the paper, $Z, Z_1, Z_2$ will always denote independent standard Gaussian variables.

### 2.2 LIMITING BEHAVIOUR OF THE VARIANCE AND COVARIANCE OPERATORS

We analyze here the limiting behaviour of $q_a^L$ and $c_{a,b}^L$ as the network depth $L$ goes to infinity under the assumption that $\phi$ has a second derivative at least in the distribution sense[1]. From now onwards, we will also assume without loss of generality that $c_{ab}^1 \geq 0$ (similar results can be obtained straightforwardly when $c_{ab}^1 \leq 0$). We first need to define the *Domains of Convergence* associated with an activation function $\phi$.

---

[1]ReLU admits a Dirac mass in 0 as second derivative and so is covered by our developments.

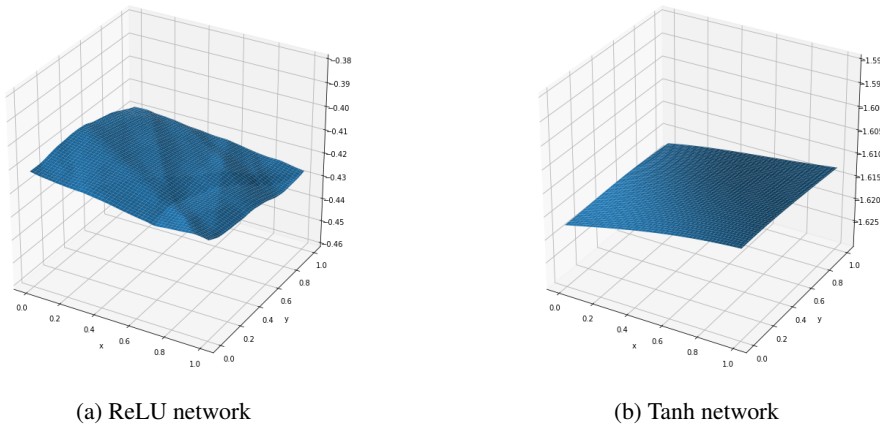

(a) ReLU network                    (b) Tanh network

Figure 1: Two draws of outputs for ReLU and Tanh networks with $(\sigma_b, \sigma_w) = (1, 1) \in D_{\phi,var} \cap D_{\phi,corr}$. The output functions are almost constant.

**Definition 1.** *Let $\phi$ be an activation function, $(\sigma_b, \sigma_w) \in (\mathbb{R}^+)^2$.*
*(i) $(\sigma_b, \sigma_w)$ is in $D_{\phi,var}$ (domain of convergence for the variance) if there exists $K > 0$, $q \geq 0$ such that for any input $a$ with $q_a^1 \leq K$, $\lim_{l \to \infty} q_a^l = q$. We denote by $K_{\phi,var}(\sigma_b, \sigma_w)$ the maximal $K$ satisfying this condition.*
*(ii) $(\sigma_b, \sigma_w)$ is in $D_{\phi,corr}$ (domain of convergence for the correlation) if there exists $K > 0$ such that for any two inputs $a, b$ with $q_a^1, q_b^1 \leq K$, $\lim_{l \to \infty} c_{ab}^l = 1$. We denote by $K_{\phi,corr}(\sigma_b, \sigma_w)$ the maximal $K$ satisfying this condition.*

*Remark :* Typically, $q$ in Definition 1 is a fixed point of the variance function defined in equation 2. Therefore, it is easy to see that for any $(\sigma_b, \sigma_w)$ such that $F$ is increasing and admits at least one fixed point, we have $K_{\phi,corr}(\sigma_b, \sigma_w) \geq q$ where $q$ is the minimal fixed point; i.e. $q := \min\{x : F(x) = x\}$. Thus, if we re-scale the input data to have $q_a^1 \leq q$, the variance $q_a^l$ converges to $q$. We can also re-scale the variance $\sigma_w$ of the first layer (only) to assume that $q_a^1 \leq q$ for all inputs $a$.

The next result gives sufficient conditions on $(\sigma_b, \sigma_w)$ to be in the domains of convergence of $\phi$.

**Proposition 1.** *Let $M_\phi := \sup_{x \geq 0} \mathbb{E}[|\phi'^2(xZ) + \phi''(xZ)\phi(xZ)|]$. Assume $M_\phi < \infty$, then for $\sigma_w^2 < \frac{1}{M_\phi}$ and any $\sigma_b$, we have $(\sigma_b, \sigma_w) \in D_{\phi,var}$ and $K_{\phi,var}(\sigma_b, \sigma_w) = \infty$*

*Let $C_{\phi,\delta} := \sup_{x,y \geq 0, |x-y| \leq \delta, c \in [0,1]} \mathbb{E}[|\phi'(xZ_1)\phi'(y(cZ_1 + \sqrt{1-c^2}Z_2))|]$. Assume $C_{\phi,\delta} < \infty$ for some $\delta > 0$, then for $\sigma_w^2 < \min(\frac{1}{M_\phi}, \frac{1}{C_\phi})$ and any $\sigma_b$, we have $(\sigma_b, \sigma_w) \in D_{\phi,var} \cap D_{\phi,corr}$ and $K_{\phi,var}(\sigma_b, \sigma_w) = K_{\phi,corr}(\sigma_b, \sigma_w) = \infty$.*

The proof of Proposition 1 is straightforward. We prove that $\sup F'(x) = \sigma_w^2 M_\phi$ and then apply the Banach fixed point theorem; similar ideas are used for $C_{\phi,\delta}$.

*Example :* For ReLU activation function, we have $M_{ReLU} = 2$ and $C_{ReLU,\delta} \leq 1$ for any $\delta > 0$.

In the domain of convergence $D_{\phi,var} \cap D_{\phi,corr}$, for all $a, b \in \mathbb{R}^d$, $y_i^\infty(a) = y_i^\infty(b)$ almost surely and the outputs of the network are constant functions. Figure 1 illustrates this behaviour for $d = 2$ for ReLU and Tanh using a network of depth $L = 10$ with $N_l = 100$ neurons per layer. The draws of outputs of these networks are indeed almost constant.

To refine this convergence analysis, Schoenholz et al. (2017) established the existence of $\epsilon_q$ and $\epsilon_c$ such that $|q_a^l - q| \sim e^{-l/\epsilon_q}$ and $|c_{ab}^l - 1| \sim e^{-l/\epsilon_c}$ when fixed points exist. The quantities $\epsilon_q$ and $\epsilon_c$ are called 'depth scales' since they represent the depth to which the variance and correlation can propagate without being exponentially close to their limits. More precisely, if we write $\chi_1 = \sigma_w^2 \mathbb{E}[\phi'(\sqrt{q}Z)^2]$ and $\alpha = \chi_1 + \sigma_w^2 \mathbb{E}[\phi''(\sqrt{q}Z)\phi(\sqrt{q}Z)]$ then the depth scales are given by $\epsilon_r = -\log(\alpha)^{-1}$ and $\epsilon_c = -\log(\chi_1)^{-1}$. The equation $\chi_1 = 1$ corresponds to an infinite depth scale of the correlation. It is called the edge of chaos as it separates two phases: an ordered phase where the correlation

converges to 1 if $\chi_1 < 1$ and a chaotic phase where $\chi_1 > 1$ and the correlations do not converge to 1. In this chaotic regime, it has been observed in Schoenholz et al. (2017) that the correlations converge to some random value $c < 1$ when $\phi(x) = \text{Tanh}(x)$ and that $c$ is independent of the correlation between the inputs. This means that very close inputs (in terms of correlation) lead to very different outputs. Therefore, in the chaotic phase, the output function of the neural network is non-continuous everywhere.

**Definition 2.** *For $(\sigma_b, \sigma_w) \in D_{\phi, var}$, let $q$ be the limiting variance[2]. The Edge of Chaos, hereafter EOC, is the set of values of $(\sigma_b, \sigma_w)$ satisfying $\chi_1 = \sigma_w^2 \mathbb{E}[\phi'(\sqrt{q}Z)^2] = 1$.*

To further study the EOC regime, the next lemma introduces a function $f$ called the 'correlation function' simplifying the analysis of the correlations. It states that the correlations have the same asymptotic behaviour as the time-homogeneous dynamical system $c_{ab}^{l+1} = f(c_{ab}^l)$.

**Lemma 1.** *Let $(\sigma_b, \sigma_w) \in D_{\phi, var} \cap D_{\phi, corr}$ such that $q > 0$, $a, b \in \mathbb{R}^d$ and $\phi$ an activation function such that $\sup_{x \in S} \mathbb{E}[\phi(xZ)^2] < \infty$ for all compact sets $S$. Define $f_l$ by $c_{ab}^{l+1} = f_l(c_{ab}^l)$ and $f$ by $f(x) = \frac{\sigma_b^2 + \sigma_w^2 \mathbb{E}[\phi(\sqrt{q}Z_1)\phi(\sqrt{q}(xZ_1 + \sqrt{1-x^2}Z_2))]}{q}$. Then $\lim_{l \to \infty} \sup_{x \in [0,1]} |f_l(x) - f(x)| = 0$.*

The condition on $\phi$ in Lemma 1 is violated only by activation functions with exponential growth (which are not used in practice), so from now onwards, we use this approximation in our analysis. Note that being on the EOC is equivalent to $(\sigma_b, \sigma_w)$ satisfying $f'(1) = 1$. In the next section, we analyze this phase transition carefully for a large class of activation functions.

## 3 EDGE OF CHAOS

To illustrate the effect of the initialization on the EOC, we plot in Figure 2 the output of a ReLU neural network with 20 layers and 100 neurons per layer with parameters $(\sigma_b^2, \sigma_w^2) = (0, 2)$ (as we will see later EOC $= \{(0, \sqrt{2})\}$ for ReLU). Unlike the output in Figure 1, this output displays much more variability. However, we will prove here that the correlations still converges to 1 even in the EOC regime, albeit at a slower rate.

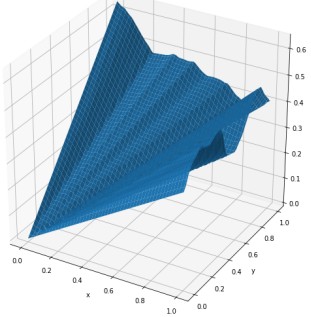

Figure 2: A draw from the output function of a ReLu network with 20 layers, 100 neurons per layer, $(\sigma_b^2, \sigma_w^2) = (0, 2)$ (edge of chaos)

### 3.1 RELU-LIKE ACTIVATION FUNCTIONS

We consider activation functions $\phi$ of the form: $\phi(x) = \lambda x$ if $x > 0$ and $\phi(x) = \beta x$ if $x \leq 0$. ReLU corresponds to $\lambda = 1$ and $\beta = 0$. For this class of activation functions, we see (Proposition 2) that the variance is unchanged ($q_a^l = q_a^1$) on the EOC, so that $q$ does not formally exist in the sense that the limit of $q_a^l$ depends on $a$. However, this does not impact the analysis of the correlations.

**Proposition 2.** *Let $\phi$ be a ReLU-like function with $\lambda$ and $\beta$ defined above. Then for any $\sigma_w < \sqrt{\frac{2}{\lambda^2 + \beta^2}}$ and $\sigma_b \geq 0$, we have $(\sigma_b, \sigma_w) \in D_{\phi, var}$ with $K_{\phi, var}(\sigma_b, \sigma_w) = \infty$. Moreover EOC $= \{(0, \frac{1}{\sqrt{\mathbb{E}[\phi'(Z)^2]}})\}$ and, on the EOC, $F(x) = x$ for any $x \geq 0$.*

This class of activation functions has the interesting property of preserving the variance across layers when the network is initialized on the EOC. However, we show in Proposition 3 below that, even in the EOC regime, the correlations converge to 1 but at a slower rate. We only present the result for ReLU but the generalization to the whole class is straightforward.

**Example : ReLU**: The EOC is reduced to the singleton $(\sigma_b^2, \sigma_w^2) = (0, 2)$, which means we should initialize a ReLU network with the parameters $(\sigma_b^2, \sigma_w^2) = (0, 2)$. This result coincides with the

---

[2]The limiting variance is a function of $(\sigma_b, \sigma_w)$ but we do not emphasize it notationally.

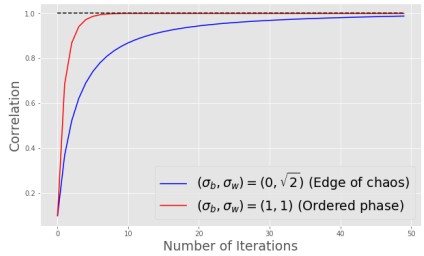 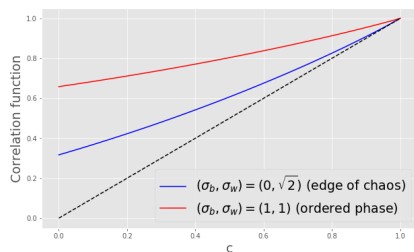

(a) Convergence of correlation $c_{ab}^l$ to 1 with $c_{ab}^0 = 0.1$

(b) Correlation function $f$

Figure 3: Impact of the initialization on the EOC for a ReLU network

recommendation of He et al. (2015) whose objective was to make the variance constant as the input propagates but did not analyze the propagation of the correlations. Klambauer et al. (2017) also performed a similar analysis by using the "Scaled Exponential Linear Unit" activation (SELU) that makes it possible to center the mean and normalize the variance of the post-activation $\phi(y)$. The propagation of the correlation was not discussed therein either. In the next result, we present the correlation function corresponding to ReLU networks. This was first obtained in Cho & Saul (2009). We present an alternative derivation of this result and further show that the correlations converge to 1 at a polynomial rate of $1/l^2$ instead of an exponential rate.

**Proposition 3** (ReLU kernel). *Consider a ReLU network with parameters $(\sigma_b^2, \sigma_w^2) = (0, 2)$ on the EOC. We have*
*(i) for $x \in [0, 1]$, $f(x) = \frac{1}{\pi} x \arcsin(x) + \frac{1}{\pi}\sqrt{1 - x^2} + \frac{1}{2}x$,*
*ii) for any $(a, b)$, $\lim_{l \to \infty} c_{ab}^l = 1$ and $1 - c_{ab}^l \sim \frac{9\pi^2}{2l^2}$ as $l \to \infty$.*

Figure 3 displays the correlation function $f$ with two different sets of parameters $(\sigma_b, \sigma_w)$. The red graph corresponds to the EOC $(\sigma_b^2, \sigma_w^2) = (0, 2)$, and the blue one corresponds to an ordered phase $(\sigma_b, \sigma_w) = (1, 1)$. In unreported experiments, we observed that numerical convergence towards 1 for $l \geq 50$ on the EOC. As the variance $q_a^l$ is preserved by the network ($q_a^l = q_a^1 = 2\|a\|^2/d$) and the correlations $c_{ab}^l$ converge to 1 as $l$ increases, the output function is of the form $C \cdot \|a\|$ for a constant $C$ (notice that in Figure 2, we start observing this effect for depth 20).

## 3.2 A BETTER CLASS OF ACTIVATION FUNCTIONS

We now introduce a set of sufficient conditions for activation functions which ensures that it is then possible to tune $(\sigma_b, \sigma_w)$ to slow the convergence of the correlations to 1. This is achieved by making the correlation function $f$ sufficiently close to the identity function.

**Proposition 4** (Main Result). *Let $\phi$ be an activation function. Assume that*
*(i) $\phi(0) = 0$, and $\phi$ has right and left derivatives in zero and $\phi'(0^+) \neq 0$ or $\phi'(0^-) \neq 0$, and there exists $k > 0$ such that $\left|\frac{\phi(x)}{x}\right| \leq k$.*
*(ii) There exists $A > 0$ such that for any $\sigma_b \in [0, A]$, there exists $\sigma_w > 0$ such that $(\sigma_b, \sigma_w) \in EOC$.*
*(iii) For any $\sigma_b \in [0, A]$, the function $F$ with parameters $(\sigma_b, \sigma_w) \in EOC$ is non-decreasing and $\lim_{\sigma_b \to 0} q = 0$ where $q$ is the minimal fixed point of $F$, $q := \inf\{x : F(x) = x\}$.*
*(iv) For any $\sigma_b \in [0, A]$, the correlation function $f$ with parameters $(\sigma_b, \sigma_w) \in EOC$ introduced in Lemma 1 is convex.*

*Then, for any $\sigma_b \in [0, A]$, we have $K_{\phi, var}(\sigma_b, \sigma_w) \geq q$, and*

$$\lim_{\substack{\sigma_b \to 0 \\ (\sigma_b, \sigma_w) \in EOC}} \sup_{x \in [0,1]} |f(x) - x| = 0.$$

Note that ReLU does not satisfy the condition $(ii)$ since the EOC in this case is the singleton $(\sigma_b^2, \sigma_w^2) = (0, 2)$. The result of Proposition 4 states that we can make $f(x)$ close to $x$ by considering $\sigma_b \to 0$. However, this is under condition $(iii)$ which states that $\lim_{\sigma_b \to 0} q = 0$. Therefore, practically, we cannot take $\sigma_b$ too small. One might wonder whether condition $(iii)$ is necessary for this result to hold. The next lemma shows that removing this condition results in a useless class of activation functions.

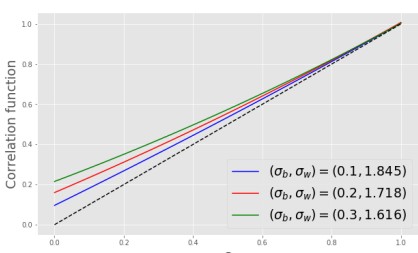
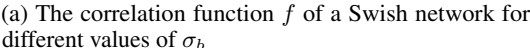

(a) The correlation function $f$ of a Swish network for different values of $\sigma_b$

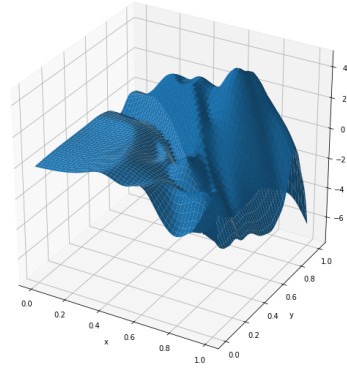

(b) A draw from the output function of a Swish network with depth 30 and width 100 on the edge of chaos for $\sigma_b = 0.2$

Figure 4: Correlation function and a draw of the output for a Swish network

**Lemma 2.** *Under the conditions of Proposition 4, the only change being $\lim_{\sigma_b \to 0} q > 0$, the result of Proposition 4 holds if only if the activation function is linear.*

The next proposition gives sufficient conditions for bounded activation functions to satisfy all the conditions of Proposition 4.

**Proposition 5.** *Let $\phi$ be a bounded function such that $\phi(0) = 0$, $\phi'(0) > 0$, $\phi'(x) \geq 0$, $\phi(-x) = -\phi(x)$, $x\phi(x) > 0$ and $x\phi''(x) < 0$ for $x \neq 0$, and $\phi$ satisfies (ii) in Proposition 4. Then, $\phi$ satisfies all the conditions of Proposition 4.*

The conditions in Proposition 5 are easy to verify and are, for example, satisfied by Tanh and Arctan. We can also replace the assumption "$\phi$ satisfies (ii) in Proposition 4" by a sufficient condition (see Proposition 7 in the Supplementary Material). Tanh-like activation functions provide better information flow in deep networks compared to ReLU-like functions. However, these functions suffer from the vanishing gradient problem during back-propagation; see, e.g., Pascanu et al. (2013) and Kolen & Kremer (2001). Thus, an activation function that satisfies the conditions of Proposition 4 (in order to have a good 'information flow') and does not suffer from the vanishing gradient issue is expected to perform better than ReLU. Swish is a good candidate.

**Proposition 6.** *The Swish activation function $\phi_{swish}(x) = x \cdot sigmoid(x) = \frac{x}{1+e^{-x}}$ satisfies all the conditions of Proposition 4.*

It is clear that Swish does not suffer from the vanishing gradient problem as it has a gradient close to 1 for large inputs like ReLU. Figure 4 (a) displays $f$ for Swish for different values of $\sigma_b$. We see that $f$ is indeed approaching the identity function when $\sigma_b$ is small, preventing the correlations from converging to 1. Figure 4(b) displays a draw of the output of a neural network of depth 30 and width 100 with Swish activation, and $\sigma_b = 0.2$. The outputs displays much more variability than the ones of the ReLU network with the same architecture. We present in Table 1 some values of $(\sigma_b, \sigma_w)$ on the EOC as well as the corresponding limiting variance for Swish. As condition $(iii)$ of Proposition 4 is satisfied, the limiting variance $q$ decreases with $\sigma_b$.

Table 1: Values of $(\sigma_b, \sigma_w)$ on the EOC and limiting variance $q$ for Swish

| $\sigma_b$ | 0.1 | 0.2 | 0.3 | 0.4 | 0.5 |
|---|---|---|---|---|---|
| $\sigma_w$ | 1.845 | 1.718 | 1.616 | 1.537 | 1.485 |
| $q$ | 0.14 | 0.44 | 0.61 | 1.01 | 2.13 |

Other activation functions that have been shown to outperform empirically ReLU such as ELU (Clevert et al. (2016)), SELU (Klambauer et al. (2017)) and Softplus also satisfy the conditions of Proposition 4 (see Supplementary Material for ELU). The comparison of activation functions satisfying the conditions of Proposition 4 remains an open question.

## 4    EXPERIMENTAL RESULTS

We demonstrate empirically our results on the MNIST dataset. In all the figures below, we compare the learning speed (test accuracy with respect to the number of epochs/iterations) for different activation functions and initialization parameters. We use the Adam optimizer with learning rate lr = 0.001. The Python code to reproduce all the experiments will be made available on-line.

**Initialization on the Edge of Chaos** We initialize randomly the deep network by sampling $W_{ij}^l \overset{iid}{\sim} \mathcal{N}(0, \sigma_w^2/N_{l-1})$ and $B_i^l \overset{iid}{\sim} \mathcal{N}(0, \sigma_b^2)$. In Figure 5, we compare the learning speed of a Swish network for different choices of random initialization. Any initialization other than on the edge of chaos results in the optimization algorithm being stuck eventually at a very poor test accuracy of $\sim 0.1$ as the depth $L$ increases (equivalent to selecting the output uniformly at random). To understand what is happening in this case, let us recall how the optimization algorithm works. Let $\{(X_i, Y_i), 1 \leq i \leq N\}$ be the MNIST dataset. The loss we optimize is given by $\mathcal{L}(w, b) = \sum_{i=1}^{N} \ell(y^L(X_i), Y_i)/N$ where $y^L(x)$ is the output of the network, and $\ell$ is the categorical cross-entropy loss. In the ordered phase, we know that the output converges exponentially to a fixed value (same value for all $X_i$), thus a small change in $w$ and $b$ will not change significantly the value of the loss function, therefore the gradient is approximately zero and the gradient descent algorithm will be stuck around the initial value.

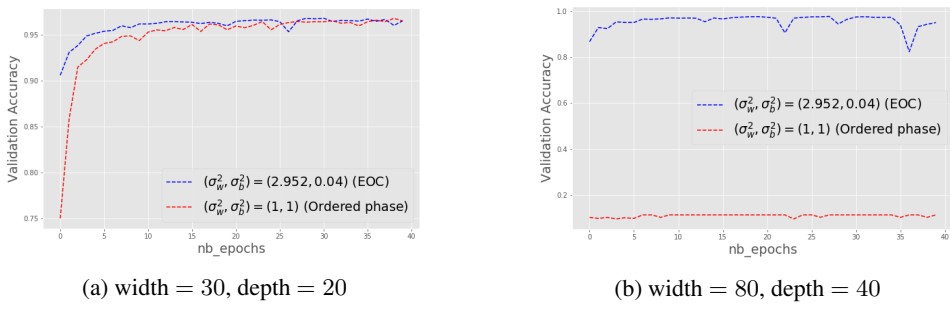

(a) width = 30, depth = 20    (b) width = 80, depth = 40

Figure 5: Impact of the initialization on the edge of chaos for Swish network

**ReLU versus Tanh** We proved in Section 3.2 that the Tanh activation guarantees better information propagation through the network when initialized on the EOC. However, Tanh suffers

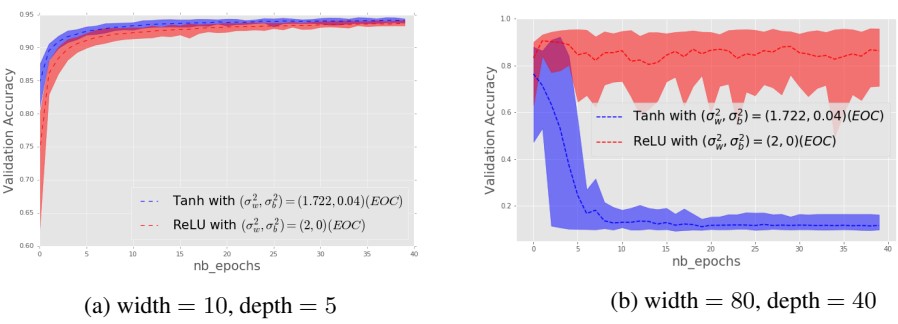

(a) width = 10, depth = 5    (b) width = 80, depth = 40

Figure 6: Comparaison of ReLu and Tanh learning curves for different widths and depths

from the vanishing gradient problem. Consequently, we expect Tanh to perform better than ReLU for shallow networks as opposed to deep networks, where the problem of the vanishing gradient is not encountered. Numerical results confirm this fact. Figure 6 shows curves of validation accuracy with confidence interval 90% (30 simulations). For depth 5, the learning algorithm converges faster for Tanh compared to ReLu. However, for deeper networks ($L \geq 40$), Tanh is stuck at a very low test accuracy, this is due to the fact that a lot of parameters remain essentially unchanged because the gradient is very small.

**ReLU versus Swish** As established in Section 3.2, Swish, like Tanh, propagates the information better than ReLU and, contrary to Tanh, it does not suffer from the vanishing gradient problem. Hence our results suggest that Swish should perform better than ReLU, especially for deep architectures. Numerical results confirm this fact. Figure 7 shows curves of validation accuracy with confidence interval 90% (30 simulations). Swish performs clearly better than ReLU especially for depth 40. A comparative study of final accuracy is shown in Table 2. We observe a clear advantage for Swish, especially for large depths. Additional simulations results on diverse datasets demonstrating better performance of Swish over many other activation functions can be found in Ramachandran et al. (2017) (Notice that these authors have already implemented Swish in Tensorflow).

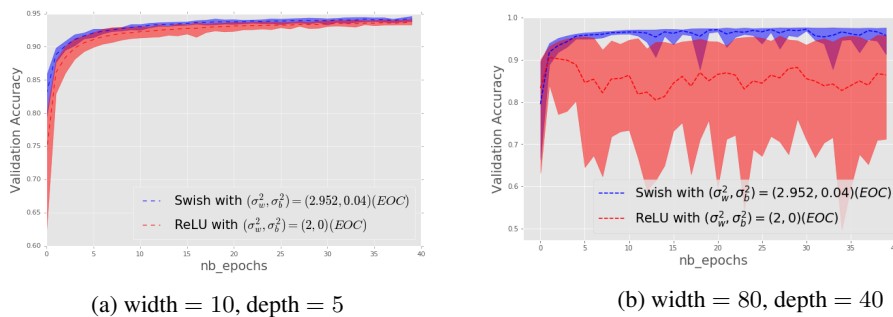

(a) width = 10, depth = 5  (b) width = 80, depth = 40

Figure 7: Convergence across iterations of the learning algorithm for ReLU and Swish networks

Table 2: Accuracy on test set for different values of (width, depth)

|       | (10,5) | (20,10) | (40,30) | (60,40) |
|-------|--------|---------|---------|---------|
| ReLU  | 94.01  | 96.01   | 96.51   | 91.45   |
| Swish | **94.46** | **96.34** | **97.09** | **97.14** |

## 5 CONCLUSION AND DISCUSSION

We have complemented here the analysis of Schoenholz et al. (2017) which shows that initializing networks on the EOC provides a better propagation of information across layers. In the ReLU case, such an initialization corresponds to the popular approach proposed in He et al. (2015). However, even on the EOC, the correlations still converge to 1 at a polynomial rate for ReLU networks. We have obtained a set of sufficient conditions for activation functions which further improve information propagation when the parameters $(\sigma_b, \sigma_w)$ are on the EOC. The Tanh activation satisfied those conditions but, more interestingly, other functions which do not suffer from the vanishing/exploding gradient problems also verify them. This includes the Swish function used in Hendrycks & Gimpel (2016), Elfwing et al. (2017) and promoted in Ramachandran et al. (2017) but also ELU Clevert et al. (2016).

Our results have also interesting implications for Bayesian neural networks which have received renewed attention lately; see, e.g., Hernandez-Lobato & Adams (2015) and Lee et al. (2018). They show that if one assigns i.i.d. Gaussian prior distributions to the weights and biases, the resulting prior distribution will be concentrated on close to constant functions even on the EOC for ReLU-like activation functions. To obtain much richer priors, our results indicate that we need to select not only parameters $(\sigma_b, \sigma_w)$ on the EOC but also an activation function satisfying Proposition 4.

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

## A    PROOFS

We provide in the supplementary material the proofs of the propositions presented in the main document, and we give additive theoretical and experimental results. For the sake of clarity we recall the propositions before giving their proofs.

### A.1    CONVERGENCE TO THE FIXED POINT: PROPOSITION 1

**Proposition 1.** *Let $M_\phi := \sup_{x \geq 0} \mathbb{E}[|\phi'^2(xZ) + \phi''(xZ)\phi(xZ)|]$. Suppose $M_\phi < \infty$, then for $\sigma_w^2 < \frac{1}{M_\phi}$ and any $\sigma_b$, we have $(\sigma_b, \sigma_w) \in D_{\phi,var}$ and $K_{\phi,var}(\sigma_b, \sigma_w) = \infty$*

*Moreover, let $C_{\phi,\delta} := \sup_{x,y \geq 0, |x-y| \leq \delta, c \in [0,1]} \mathbb{E}[|\phi'(xZ_1)\phi'(y(cZ_1 + \sqrt{1-c^2}Z_2))|]$. Suppose $C_{\phi,\delta} < \infty$ for some positive $\delta$, then for $\sigma_w^2 < \min(\frac{1}{M_\phi}, \frac{1}{C_\phi})$ and any $\sigma_b$, we have $(\sigma_b, \sigma_w) \in D_{\phi,var} \cap D_{\phi,corr}$ and $K_{\phi,var}(\sigma_b, \sigma_w) = K_{\phi,corr}(\sigma_b, \sigma_w) = \infty$.*

*Proof.* To abbreviate the notation, we use $q^l := q_a^l$ for some fixed input a.

**Convergence of the variances:** We first consider the asymptotic behaviour of $q^l = q_a^l$. Recall that $q^l = F(q^{l-1})$ where,
$$F(x) = \sigma_b^2 + \sigma_w^2 \mathbb{E}[\phi(\sqrt{x}Z)^2].$$

The first derivative of this function is given by:
$$F'(x) = \sigma_w^2 \mathbb{E}[\frac{Z}{\sqrt{x}}\phi'(\sqrt{x}Z)\phi(\sqrt{x}Z)] = \sigma_w^2 \mathbb{E}[\phi'(\sqrt{x}Z)^2 + \phi''(\sqrt{x}Z)\phi(\sqrt{x}Z)] \tag{3}$$

where we used Gaussian integration by parts $\mathbb{E}[ZG(Z)] = \mathbb{E}[G'(Z)]$, an identify satisfied by any function $G$ such that $\mathbb{E}[|G'(Z)|] < \infty$.

Using the condition on $\phi$, we see that for $\sigma_w^2 < \frac{1}{M_\phi}$, the function $F$ is a contraction mapping, and the Banach fixed-point theorem guarantees the existence of a unique fixed point $q$ of $F$, with $\lim_{l \to +\infty} q^l = q$. Note that this fixed point depends only on $F$, therefore, this is true for any input $a$, and $K_{\phi,var}(\sigma_b, \sigma_w) = \infty$.

**Convergence of the covariances:** Since $M_\phi < \infty$, then for all $a, b \in \mathbb{R}^d$ there exists $l_0$ such that, for all $l > l_0$, $|\sqrt{q_a^l} - \sqrt{q_b^l}| < \delta$. Let $l > l_0$, using Gaussian integration by parts, we have

$$\frac{dc_{ab}^{l+1}}{dc_{ab}^l} = \sigma_w^2 \mathbb{E}[|\phi'(\sqrt{q_a^l}Z_1)\phi'(\sqrt{q_b^l}(c_{ab}^l Z_1 + \sqrt{1-(c_{ab}^l)^2}Z_2)|].$$

We cannot use the Banach fixed point theorem directly because the integrated function here depends on $l$ through $q^l$. For ease of notation, we write $c^l := c_{ab}^l$, we have

$$|c^{l+1} - c^l| = |\int_{c^{l-1}}^{c^l} \frac{dc^{l+1}}{dc^l}(x)dx| \leq \sigma_w^2 C_\phi |c^l - c^{l-1}|.$$

Therefore, for $\sigma_w^2 < \min(\frac{1}{M_\phi}, \frac{1}{C_\phi})$, $c^l$ is a Cauchy sequence and it converges to a limit $c \in [0,1]$. At the limit
$$c = f(c) = \frac{\sigma_b^2 + \sigma_w^2 \mathbb{E}[\phi(\sqrt{q}z_1)\phi(\sqrt{q}(cz_1 + \sqrt{1-c^2}z_2)))]}{q},$$

The derivative of this function is given by
$$f'(x) = \sigma_w^2 \mathbb{E}[\phi'(\sqrt{q}Z_1)\phi'(\sqrt{q}(xZ_1 + \sqrt{1-x}Z_2))]$$

By assumption on $\phi$ and the choice of $\sigma_w$, we have $\sup_x |f'(x)| < 1$, so that $f$ is a contraction, and has a unique fixed point. Since $f(1) = 1$, $c = 1$. The above result is true for any $a, b$, therefore, $K_{\phi,var}(\sigma_b, \sigma_w) = K_{\phi,corr}(\sigma_b, \sigma_w) = \infty$. □

As an illustration we plot in Figure 12 the variance for three different inputs with $(\sigma_b, \sigma_w) = (1, 1)$, as a function of the layer $l$. In this example, the convergence for Tanh is faster than that of ReLU.

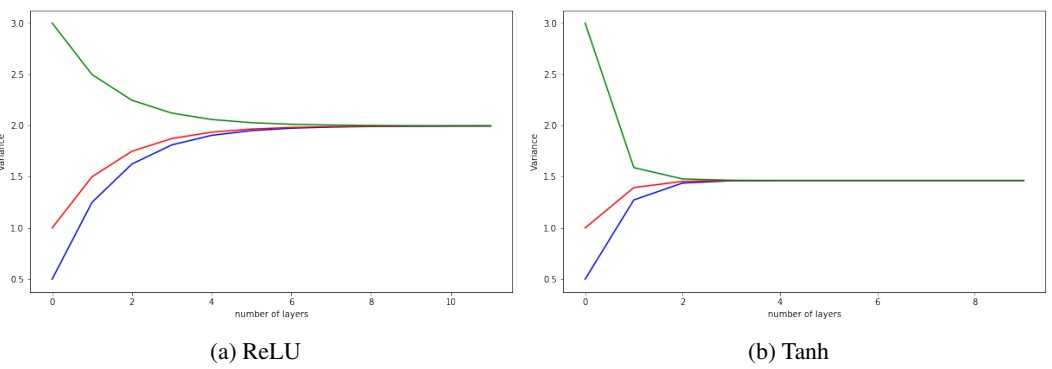

|           |           |
| :-------: | :-------: |
| (a) ReLU  | (b) Tanh  |

Figure 8: Convergence of the variance for three different inputs with $(\sigma_b, \sigma_w) = (1, 1)$

**Lemma 1.** *Let $(\sigma_b, \sigma_w) \in D_{\phi,var} \cap D_{\phi,corr}$ such that $q > 0$, $a, b \in \mathbb{R}^d$ and $\phi$ an activation function such that $\sup_{x \in K} \mathbb{E}[\phi(xZ)^2] < \infty$ for all compact sets $K$. Define $f_l$ by $c_{a,b}^{l+1} = f_l(c_{a,b}^l)$ and $f$ by $f(x) = \frac{\sigma_b^2 + \sigma_w^2 \mathbb{E}[\phi(\sqrt{q}Z_1)\phi(\sqrt{q}(xZ_1 + \sqrt{1-x^2}Z_2))]}{q}$. Then $\lim_{l \to \infty} \sup_{x \in [0,1]} |f_l(x) - f(x)| = 0$.*

*Proof.* For $x \in [0, 1]$, we have

$$f_l(x) - f(x) = (\frac{1}{\sqrt{q_a^l q_b^l}} - \frac{1}{q})(\sigma_b^2 + \sigma_w^2 \mathbb{E}[\phi(\sqrt{q_a^l}Z_1)\phi(\sqrt{q_b^l}u_2(x))])$$
$$+ \frac{\sigma_w^2}{q}(\mathbb{E}[\phi(\sqrt{q_a^l}Z_1)\phi(\sqrt{q_b^l}u_2(x))] - \mathbb{E}[\phi(\sqrt{q}Z_1)\phi(\sqrt{q}u_2(x))]),$$

where $u_2(x) := xZ_1 + \sqrt{1 - x^2}Z_2$. The first term goes to zero uniformly in $x$ using the condition on $\phi$ and Cauchy-Schwartz inequality. As for the second term, it can be written as

$$\mathbb{E}[(\phi(\sqrt{q_a^l}Z_1) - \phi(\sqrt{q}Z_1))\phi(\sqrt{q_b^l}u_2(x))] + \mathbb{E}[\phi(\sqrt{q}Z_1)(\phi(\sqrt{q_b^l}u_2(x)) - \phi(\sqrt{q}u_2(x)))]$$

again, using Cauchy-Schwartz and the condition on $\phi$, both terms can be controlled uniformly in $x$ by an integrable upper bound. We conclude using the Dominated convergence. $\square$

### A.2   RESULTS FOR RELU-LIKE ACTIVATION FUNCTIONS: PROOF OF PROPOSITIONS 2 AND 3

**Proposition 2.** *Let $\phi$ be a ReLU-like function with $\lambda$ and $\beta$ defined above. Then for any $\sigma_w < \sqrt{\frac{2}{\lambda^2 + \beta^2}}$ and $\sigma_b \geq 0$, we have $(\sigma_b, \sigma_w) \in D_{\phi,var}$ with $K_{\phi,var}(\sigma_b, \sigma_w) = \infty$. Moreover $EOC = \{(0, \frac{1}{\sqrt{\mathbb{E}[\phi'(Z)^2]}})\}$ and, on the EOC, $F(x) = x$ for any $x \geq 0$.*

*Proof.* We write $q^l = q_a^l$ throughout the proof. Note first that the variance satisfies the recursion:

$$q^{l+1} = \sigma_b^2 + \sigma_w^2 \mathbb{E}[\phi(Z)^2]q^l = \sigma_b^2 + \sigma_w^2 \frac{\lambda^2 + \beta^2}{2}q^l. \tag{4}$$

For all $\sigma_w < \sqrt{\frac{2}{\lambda^2 + \beta^2}}$, $q = \sigma_b^2\left(1 - \sigma_w^2(\lambda^2 + \beta^2)/2\right)^{-1}$ is a fixed point. This is true for any input, therefore $K_{\phi,var}(\sigma_b, \sigma_w) = \infty$ and (i) is proved.

Now, the EOC equation is given by $\chi_1 = \sigma_w^2 \mathbb{E}[\phi'(Z)^2] = \sigma_w^2 \frac{\lambda^2 + \beta^2}{2}$. Therefore, $\sigma_w^2 = \frac{2}{\lambda^2 + \beta^2}$. Replacing $\sigma_w^2$ by its critical value in equation 4 yields

$$q^{l+1} = \sigma_b^2 + q^l.$$

Thus $q = \sigma_b^2 + q$ if and only if $\sigma_b = 0$, otherwise $q^l$ diverges to infinity. So the frontier is reduced to a single point $(\sigma_b^2, \sigma_w^2) = (0, \mathbb{E}[\phi'(Z)^2]^{-1})$, and the variance does not depend on $l$.

$\square$

**Proposition 3** (ReLU kernel). *Consider a ReLU network with parameters $(\sigma_b^2, \sigma_w^2) = (0, 2)$ on the EOC. We have*
*(i) for $x \in [0, 1]$, $f(x) = \frac{1}{\pi} x \arcsin(x) + \frac{1}{\pi}\sqrt{1 - x^2} + \frac{1}{2}x$,*
*ii) for any $(a, b)$, $\lim_{l \to \infty} c_{ab}^l = 1$ and $1 - c_{ab}^l \sim \frac{9\pi^2}{2l^2}$ as $l \to \infty$.*

*Proof.* In this case the correlation function $f$ is given by $f(x) = 2\mathbb{E}[(Z_1)_+(xZ_1 + \sqrt{1 - x^2}Z_2)_+]$ where $(x)_+ := x1_{x>0}$.

- Let $x \in [0, 1]$, note that $f$ is differentiable and satisfies,

$$f'(x) = 2\mathbb{E}[1_{Z_1 > 0}1_{xZ_1 + \sqrt{1-x^2}Z_2 > 0}],$$

  which is also differentiable. Simple algebra leads to

$$f''(x) = \frac{1}{\pi\sqrt{1 - x^2}}.$$

  Since $\arcsin'(x) = \frac{1}{\sqrt{1-x^2}}$ and $f'(0) = 1/2$,

$$f'(x) = \frac{1}{\pi}\arcsin(x) + \frac{1}{2}.$$

  We conclude using the fact that $\int \arcsin = x\arcsin + \sqrt{1 - x^2}$ and $f(1) = 1$.

- We first derive a Taylor expansion of $f$ near 1. Consider the change of variable $x = 1 - t^2$ with $t$ close to 0, then

$$\arcsin(1 - t^2) = \frac{\pi}{2} - \sqrt{2}t - \frac{\sqrt{2}}{12}t^3 + O(t^5),$$

  so that

$$\arcsin(x) = \frac{\pi}{2} - \sqrt{2}(1 - x)^{1/2} - \frac{\sqrt{2}}{12}(1 - x)^{3/2} + O((1 - x)^{5/2}),$$

  and

$$x\arcsin(x) = \frac{\pi}{2} - \sqrt{2}(1 - x)^{1/2} + \frac{11\sqrt{2}}{12}(1 - x)^{3/2} + O((1 - x)^{5/2}).$$

  Since

$$\sqrt{1 - x^2} = \sqrt{2}(1 - x)^{1/2} - \frac{\sqrt{2}}{4}(1 - x)^{3/2} + O((1 - x)^{5/2}),$$

  we obtain that

$$f(x) \underset{x \to 1-}{=} x + \frac{2\sqrt{2}}{3\pi}(1 - x)^{3/2} + O((1 - x)^{5/2}). \tag{5}$$

  Since $(f(x) - x)' = \frac{1}{\pi}(\arcsin(x) - \frac{\pi}{2}) < 0$ and $f(1) = 1$, for all $x \in [0, 1[$, $f(x) > x$. If $c^l < c^{l+1}$ then by taking the image by $f$ (which is increasing because $f' \geq 0$) we have that $c^{l+1} < c^{l+2}$, and we know that $c^1 = f(c^0) \geq c^0$, so by induction the sequence $c^l$ is increasing, and therefore it converges (because it is bounded) to the fixed point of $f$ which is 1.

  Now let $\gamma_l := 1 - c_{ab}^l$ for $a, b$ fixed. We note $s = \frac{2\sqrt{2}}{3\pi}$, from the series expansion we have that $\gamma_{l+1} = \gamma_l - s\gamma_l^{3/2} + O(\gamma_l^{5/2})$ so that

$$\gamma_{l+1}^{-1/2} = \gamma_l^{-1/2}(1 - s\gamma_l^{1/2} + O(\gamma_l^{3/2}))^{-1/2} = \gamma_l^{-1/2}(1 + \frac{s}{2}\gamma_l^{1/2} + O(\gamma_l^{3/2}))$$

$$= \gamma_l^{-1/2} + \frac{s}{2} + O(\gamma_l).$$

Thus, as $l$ goes to infinity

$$\gamma_{l+1}^{-1/2} - \gamma_l^{-1/2} \sim \frac{s}{2}$$

and by summing and equivalence of positive divergent series

$$\gamma_l^{-1/2} \sim \frac{s}{2}l,$$

which terminates the proof.

$\square$

### A.3 A BETTER CLASS OF ACTIVATION FUNCTIONS: PROOFS OF PROPOSITIONS 4, 5, 6 AND LEMMA 2

**Proposition 4** (main result). *Let $\phi$ be an activation function. Recall the definition of the variance function $F(x) := \sigma_b^2 + \sigma_w^2 \mathbb{E}[\phi(\sqrt{x}Z)^2]$. Assume that*
*(i) $\phi(0) = 0$, and $\phi$ has right and left derivatives in zero and at least one of them is different from zero ($\phi'(0^+) \neq 0$ or $\phi'(0^-) \neq 0$), and there exists $K > 0$ such that $\left|\frac{\phi(x)}{x}\right| \leq K$.*
*(ii) There exists $A > 0$ such that for any $\sigma_b \in [0, A]$, there exists $\sigma_{w,EOC} > 0$ such that $(\sigma_b, \sigma_{w,EOC}) \in EOC$.*
*(iii) For any $\sigma_b \in [0, A]$, the function $F$ with parameters $(\sigma_b, \sigma_{w,EOC})$ is non-decreasing and $\lim_{\sigma_b \to 0} q = 0$ where $q$ is the minimal fixed point of $F$, $q := \inf\{x : F(x) = x\}$).*
*(iv) For any $\sigma_b \in [0, A]$, the correlation function $f$ with parameters $(\sigma_b, \sigma_{w,EOC})$ introduced in Lemma 1 is convex.*

*Then, for any $\sigma_b \in [0, A]$, we have $K_{\phi,var}(\sigma_b, \sigma_w) \geq q$, and*

$$\lim_{\substack{\sigma_b \to 0 \\ (\sigma_b, \sigma_w) \in EOC}} \sup_{x \in [0,1]} |f(x) - x| = 0.$$

*Proof.* We first prove that $K_{\phi,var}(\sigma_b, \sigma_w) \geq q$. We assume that $\sigma_b > 0$, the case $\sigma_b = 0$ is trivial since in this case $q = 0$ (the output of the network is zero in this case).

Since $F$ is continuous and $F(0) = \sigma_b^2 > 0$, we have $x \leq F(x) \leq q$ for all $x \in [0, q]$. Using the fact that $F$ is non-decreasing for any input $a$ such that $q_a^1 \leq q$, we have $q^l$ is increasing and converges to the fixed point $q$. Therefore $K_{\phi,var}(\sigma_b, \sigma_w) \geq q$.

Now we prove that on the edge of chaos, we have

$$\lim_{\sigma_b \to 0} \frac{\sigma_b^2}{q} = 0. \tag{6}$$

The EOC equation is given by $\sigma_w^2 \mathbb{E}[\phi'(\sqrt{q}Z)^2] = 1$. By taking the limit $\sigma_b \to 0$ on the edge of chaos, and using the fact that $\lim_{\sigma_b \to 0} q = 0$, we have $\sigma_w^2 \frac{\phi'(0^+)^2 + \phi'(0^-)^2}{2} = 1$. Moreover $q$ verifies

$$\frac{F(q)}{q} = 1 = \frac{\sigma_b^2}{q} + \sigma_w^2 \mathbb{E}[(\frac{\phi(\sqrt{q}Z)}{\sqrt{q}})^2],$$

so that by taking the limit $\sigma_b \to 0$, and using the dominated convergence theorem, we have that $1 = \lim_{\sigma_b \to 0} \frac{\sigma_b^2}{q} + \sigma_w^2 \frac{\phi'(0^+)^2 + \phi'(0^-)^2}{2} = \lim_{\sigma_b \to 0} \frac{\sigma_b^2}{q} + 1$ and equation 6 holds.

Finally since $f$ is strictly convex, for all $x \in [0, 1]$ $f'(x) \leq f'(1) = 1$ if $(\sigma_b, \sigma_w) \in EOC$. Therefore $0 \leq f(x) - x \leq f(0) = \frac{\sigma_b^2}{q}$, we conclude using the fact that $\lim_{\sigma_b \to 0} \frac{\sigma_b^2}{q} = 0$.

Note however that for all $\sigma_b > 0$, if $(\sigma_b, \sigma_w) \in EOC$, for any inputs $a, b$, we have $\lim_{l \to \infty} c_{a,b}^l = 1$. Indeed, since $f$ is usually strictly convex (otherwise, $f$ would be equal to identity on at least a segment of $[0, 1]$) and $f'(1) = 1$, we have that $f$ is a contraction (because $f' \geq 0$), therefore the correlation converges to the unique fixed point of $f$ which is 1. Therefore, in most of the cases, the result of Proposition 4 should be seen as a way of slowing down the convergence of the correlation to 1. $\square$

**Lemma 2.** *Under the conditions of Proposition 4, the only change being $\lim_{\sigma_b \to 0} q > 0$, the result of Proposition 4 holds only if the activation function is linear.*

*Proof.* Using the convexity of $f$ and the result of Proposition 4, we have in the limit $\sigma_b \to 0$, $f'(0) = f'(1) = 1$, which is equivalent to $\mathbb{E}[\phi'(\sqrt{q}Z)^2] = \mathbb{E}[\phi'(\sqrt{q}Z)]^2$ which implies that $\text{var}(\phi'(\sqrt{q}Z)) = 0$. Therefore there exists a constant $a_1$ such that $\phi'(\sqrt{q}Z) = a_1$ almost surely. This implies $\phi' = a_1$ almost everywhere. □

**Proposition 5.** *Let $\phi$ be a bounded function such that $\phi(0) = 0$, $\phi'(0) > 0$, $\phi'(x) \geq 0$, $\phi(-x) = -\phi(x)$, $x\phi(x) > 0$ and $x\phi''(x) < 0$ for $x \neq 0$, and $\phi$ satisfies (ii) in Proposition 4. Then, $\phi$ satisfies all the conditions of Proposition 4.*

*Proof.* Let $\phi$ be an activation function that satisfies the conditions of Proposition 5.

(i) we have $\phi(0) = 0$ and $\phi'(0) > 0$. Since $\phi$ is bounded and $0 < \phi'(0) < \infty$, then there exists $K$ such that $\left|\frac{\phi(x)}{x}\right| \leq K$.

(ii) The condition (ii) is satisfied by assumption.

(iii) Let $\sigma_b > 0$ and $\sigma_w > 0$. Using equation 3 together with $\phi' > 0$, we have $F'(x) \geq 0$ so that F is non-decreasing. Moreover, we have $F(\mathbb{R}^+) \subset [B, C] := [\sigma_b^2, \sigma_b^2 + \sigma_w^2 M^2]$, therefore any fixed point of $F$ should be in $[B, C]$. We have $F(B) \geq B$ and $F(C) \leq C$ and F is strictly increasing, therefore, there exists a fixed point $q$ of $F$ in $[B, C]$. Now we prove that $\lim_{\sigma_b \to 0} q = 0$. Using the EOC equation, $q$ satisfies the equation $q = \sigma_b^2 + \frac{\mathbb{E}[\phi(\sqrt{q}Z)^2]}{\mathbb{E}[\phi'(\sqrt{q}Z)^2]}$. Now let's prove that the function $e(x) := x - \frac{\mathbb{E}[\phi(\sqrt{x}Z)^2]}{\mathbb{E}[\phi'(\sqrt{x}Z)^2]}$ is increasing near 0 which means it is an injection near 0, this is sufficient to conclude (because we take $q$ to be the minimal fixed point). After some calculus, we have

$$e'(x) = -\frac{\mathbb{E}[\phi''(\sqrt{x}Z)(\phi(\sqrt{x}Z)\mathbb{E}[\phi'(\sqrt{x}Z)^2] - \frac{Z}{\sqrt{x}}\phi'(\sqrt{x}Z)\mathbb{E}[\phi(\sqrt{x}Z)^2])]}{\mathbb{E}[\phi'(\sqrt{x}Z)^2]}$$

Using Taylor expansion near 0, after a detailed but unenlightening calculation the numerator is equal to $-2\phi'(0)^2\phi'''(0)^2 x\sqrt{x} + O(x^2)$, therefore the function $e$ is increasing near 0.

(iv) Finally, using the notations $U_1 := \sqrt{q}Z_1$ and $U_2(x) = \sqrt{q}(xZ_1 + \sqrt{1-x^2}Z_2)$, the first and second derivatives of the correlation function are given by

$$f'(x) = \sigma_w^2 \mathbb{E}[\phi'(U_1)\phi'(U_2(x))], \quad f''(x) = \sigma_w^2 q \mathbb{E}[\phi''(U_1)\phi''(U_2(x))]$$

where we used Gaussian integration by parts. Let $x > 0$, we have that

$$\mathbb{E}[\phi''(U_1)\phi''(U_2(x))] = \mathbb{E}[1_{\{U_1 \geq 0\}}\phi''(U_1)\phi''(U_2(x))] + \mathbb{E}[1_{\{U_1 \leq 0\}}\phi''(U_1)\phi''(U_2(x))]$$
$$= 2\mathbb{E}[1_{\{U_1 \geq 0\}}\phi''(U_1)\phi''(U_2(x))]$$

where we used the fact that $(Z_1, Z_2) = (-Z_1, -Z_2)$ (in distribution) and $\phi''(-y) = -\phi''(y)$ for any $y$.
Using $x\phi''(x) \leq 0$, we have $1_{\{u_1 \geq 0\}}\phi''(u_1) \leq 0$. We also have for all $y > 0$, $\mathbb{E}[\phi''(U_2(x))|U_1 = y] < 0$, this is a consequence of the fact that $\phi''$ is an odd function and that for $x > 0$ and $y > 0$, the mapping $z_2 \to xy + \sqrt{1-x^2}z_2$ moves the center of the Gaussian distribution to a strictly positive number, we conclude that $f''(x) > 0$ almost everywhere and assumption (iii) of Proposition 4 is verified.

□

**Proposition 6.** *The Swish activation function $\phi_{swish}(x) = x \cdot sigmoid(x) = \frac{x}{1+e^{-x}}$ satisfies all the conditions of Proposition 4.*

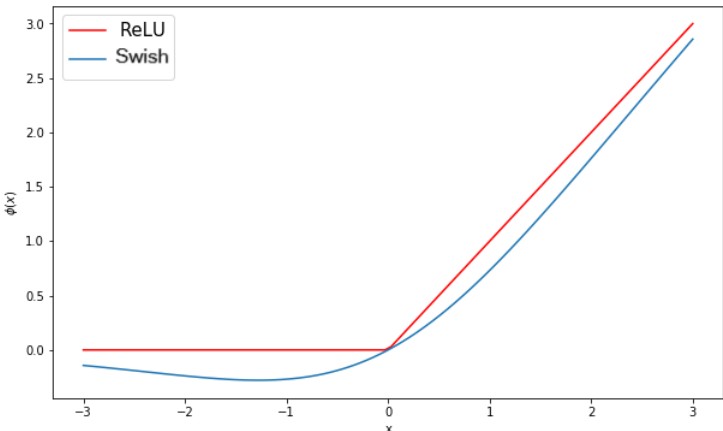

Figure 9: Graphs of ReLU and Swish

*Proof.* To abbreviate notation, we note $\phi := \phi_{Swish} = xe^x/(1 + e^x)$ and $h := e^x/(1 + e^x)$ is the Sigmoid function. This proof should be seen as a sketch of the ideas and not a rigourous proof.

- we have $\phi(0) = 0$ and $\phi'(0) = \frac{1}{2}$ and $\left|\frac{\phi(x)}{x}\right| \leq 1$

- As illustrated in Table 1 in the main text, it is easy to see numerically that (ii) is satisfied. Moreover, we observe that $\lim_{\sigma_b \to 0} q = 0$, which proves the second part of the (iii).

- Now we prove that $F' > 0$, we note $g(x) := x\phi'(x)\phi(x)$. We have

$$g(x) = x^2 \frac{(1 + e^{-x} + xe^{-x})}{(1 + e^{-x})^3}$$

Define $G$ by

$$G(x) = \begin{cases} g(x) & \text{if } x \leq 0 \\ -g(-x) & \text{if } x > 0 \end{cases}$$

so that $G(-x) = -G(x)$ for all $x \in \mathbb{R}$ and $g(x) \geq G(x)$ for all $x \leq 0$. Let $x > 0$, then

$$g(x) > G(x) \Longleftrightarrow 1 + e^{-x} + xe^{-x} \geq e^{-3x}(-1 - e^x + xe^x)$$

which holds true for any positive number $x$. We thus have $g(x) > G(x)$ for all real numbers $x$. Therefore $\mathbb{E}[g(\sqrt{x}Z)] > 0$ almost everywhere and $F' > 0$. The second part of (iii) was already proven above.

- Let $\sigma_b > 0$ and $\sigma_w > 0$ such that $q$ exists. Recall that

$$f''(x) = \sigma_w^2 q\mathbb{E}[\phi''(U_1)\phi''(U_2(x))]$$

In Figure 10, we show the graph of $\mathbb{E}[\phi''(U_1)\phi''(U_2(x))]$ for different values of $q$ (from 0.1 to 10, the darkest line being for $q = 10$). A rigorous proof can be done but is omitted here.

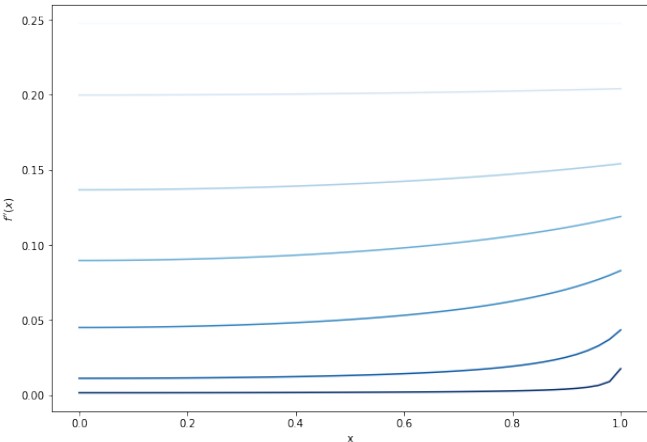

Figure 10: graphs of $\mathbb{E}[\phi''(U_1)\phi''(U_2(x))]$ for different values of $q$ (from 0.1 to 10, the darkest line corresponds to $q = 10$)

We observe that $f''$ has very small values when $q$ is large, this is a result of the fact that $\phi''$ is concentrated around 0.

*Remark :* On the edge of chaos, we have $\sigma_w^2\mathbb{E}[\phi'(\sqrt{q}Z)^2] = 1$. Recall that $F'$ can also be expressed as

$$F'(x) = \sigma_w^2(\mathbb{E}[\phi'(\sqrt{x}Z)^2] + \mathbb{E}[\phi''(\sqrt{x}Z)\phi(\sqrt{x}Z)]),$$

this yields

$$F'(q) = 1 + \sigma_w^2\mathbb{E}[\phi''(\sqrt{q}Z)\phi(\sqrt{q}Z)]. \tag{7}$$

The term $\mathbb{E}[\phi''(\sqrt{q}Z)\phi(\sqrt{q}Z)]$ is very small compared to 1 ($\sim 0.01$), therefore $F'(q) \approx 1$. Notice also that the theoretical results corresponds to the equivalent Gaussian process, which is just an approximation of the neural network. Thus, using a value of $(\sigma_b, \sigma_w)$ close to the $EOC$ should not essentially change the quality of the result. $\qquad\square$

# B    SUPPLEMENTARY THEORETICAL RESULTS

## B.1    SUFFICIENT CONDITIONS FOR BOUNDED ACTIVATION FUNCTIONS

We can replace the conditions "$\phi$ satisfies (ii)" in Proposition 5 by a sufficient condition. However, this condition is not satisfied by Tanh.

**Proposition 7.** *Let $\phi$ be a bounded function such that $\phi(0) = 0$, $\phi'(0) > 0$, $\phi'(x) \geq 0$, $\phi(-x) = -\phi(x)$, $x\phi(x) > 0$ and $x\phi''(x) < 0$ for $x \neq 0$, and $|\mathbb{E}\phi'(xZ)^2| \gtrsim |x|^{-2\beta}$ for large $x$ and some $\beta \in (0, 1)$. Then, $\phi$ satisfies all the conditions of Proposition 4.*

*Proof.* Let $\phi$ be an activation function that satisfies the conditions of Proposition 7. The proof is similar to the one of 5, we only need to show that having $|\mathbb{E}\phi'(xZ)^2| \gtrsim |x|^{-2\beta}$ for large $x$ and some $\beta \in (0, 1)$ implies that (ii) of 4 is verified.

We have that $\sigma_w^2|\mathbb{E}\phi'(\sqrt{q}Z)^2| \gtrsim \sigma_w^2 q^{-2\beta} = \sigma_w^2(\sigma_b^2 + \sigma_w^2\mathbb{E}[\phi(\sqrt{q}Z)^2])^{-\beta}$, so that we can make the term $\sigma_w^2|\mathbb{E}\phi'(\sqrt{q}Z)^2|$ take any value between 0 and $\infty$. Therefore, there exists $\sigma_w$ such that $(\sigma_b, \sigma_w) \in EOC$, and assumption (ii) of Proposition 4 holds.

$\qquad\square$

## B.2    IMPACT OF SMOOTHNESS

In the proof of Proposition 5, we used the condition on $\phi''$ (odd function) to prove that $f'' > 0$, however, in some cases when we can explicitly calculate $f$, we do not need $\phi''$ to be defined. This is the case for Hard-Tanh, which is a piecewise-linear version of Tanh. We give an explicit calculation

of $f''$ for the Hard-Tanh activation function which we note $HT$ in what follows. We compare the performance of $HT$ and Tanh based on a metric which we will define later.

$HT$ is given by

$$HT(x) = \begin{cases} -1 & \text{if } x < -1 \\ x & \text{if } -1 \leq x \leq 1 \\ 1 & \text{if } x > 1 \end{cases}$$

Recall the propagation of the variance $q^l$

$$q^{l+1} = \sigma_b^2 + \sigma_w^2 \mathbb{E}(HT(\sqrt{q^l}Z)^2)$$

where $HT$ is the Hard-Tanh function. We have

$$\mathbb{E}(HT(\sqrt{q^l}Z)^2) = \mathbb{E}(1_{Z<-\frac{1}{\sqrt{q^l}}}) + \mathbb{E}(1_{-1/\sqrt{q^l}<Z<1/\sqrt{q^l}}Z^2) + \mathbb{E}(1_{Z>1/\sqrt{q^l}})$$

$$= 1 - \frac{2}{\sqrt{q^l}}\frac{\exp(-\frac{1}{2q^l})}{\sqrt{2\pi}}$$

This yields

$$q^{l+1} = g(q^l)$$

where

$$g(x) = \sigma_b^2 + \sigma_w^2(1 - \frac{2}{\sqrt{x}}\frac{\exp(-\frac{1}{x})}{\sqrt{2\pi}})$$

- EDGE OF CHAOS :

To study the correlation behaviour, we will assume that the variance converges to $q$. We have $\mathbb{E}(HT'(\sqrt{q}Z)^2) = \mathbb{E}(1_{-\frac{1}{\sqrt{q}}<Z<\frac{1}{\sqrt{q}}}) = 2\Psi(\frac{1}{\sqrt{q}})-1$ (where $\Psi$ is the cumulative distribution function of a standard normal variable). The edge of chaos is then given by the equation $\sigma_w^2(2\Psi(\frac{1}{\sqrt{q}})-1) = 1$. We fix $\sigma_w$ to its value on the edge.

Now let $a$ and $b$ be any two inputs. We have

$$c^{l+1} = f(c^l)$$

where

$$f(x) = \frac{\sigma_b^2 + \sigma_w^2 \mathbb{E}[\phi(U_1)\phi(U_2(x))]}{q}$$

with $U_1 := \sqrt{q}Z_1$, and $U_2(x) := \sqrt{q}(xZ_1 + \sqrt{1-x^2}Z_2)$.

**Lemma 3.** *Suppose $q^l$ converges to a fixed point $q > 0$. Then,*

$$\forall x \in [0, 1], f''(x) = \frac{\sigma_w^2}{\pi\sqrt{1-x^2}}(e^{-\frac{1}{q(1+x)}} - e^{-\frac{1}{q(1-x)}})$$

*Proof.* We note $\alpha := 1/\sqrt{q}$. For $x \in [0, 1[$, we have that :

$$f'(x) = \sigma_w^2 \mathbb{E}[1_{-\alpha<Z_1<\alpha} \times 1_{-\alpha<xZ_1+\sqrt{1-x^2}Z_2<\alpha}]$$

$$= \sigma_w^2 \mathbb{E}[1_{-\alpha<Z_1<\alpha} \times (1_{-\alpha<xZ_1+\sqrt{1-x^2}Z_2} - 1_{\alpha<xZ_1+\sqrt{1-x^2}Z_2})]$$

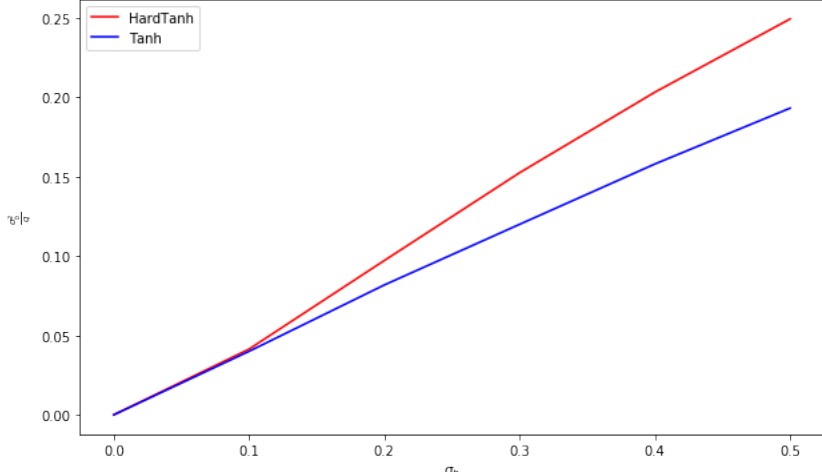

Figure 11: The correlation function on the edge of order-to-chaos for a Tanh network with small values of $\sigma_b$

We deal with the first part $f_1(x) = \sigma_w^2 \mathbb{E}[1_{-\alpha < Z_1 < \alpha} \times 1_{-\alpha < xZ_1 + \sqrt{1-x^2}Z_2}]$, we have that :

$$
\begin{aligned}
f_1'(x) &= \sigma_w^2 \mathbb{E}[1_{-\alpha < Z_1 < \alpha} \times \frac{1}{\sqrt{1-x^2}}(Z_1 - \frac{x}{\sqrt{1-x^2}}Z_2)\delta_{-\alpha = xZ_1 + \sqrt{1-x^2}Z_2}] \\
&= \frac{\sigma_w^2}{\sqrt{2\pi}} \mathbb{E}[1_{-\alpha < Z_1 < \alpha} \times \frac{1}{\sqrt{1-x^2}} \frac{Z_1 + x\alpha}{1 - x^2} \exp(-\frac{(xZ_1 + \alpha)^2}{2(1-x^2)})] \\
&= \frac{\sigma_w^2}{2\pi\sqrt{1-x^2}} \int_{-\alpha}^{\alpha} \frac{z_1 + x\alpha}{1 - x^2} \exp(-\frac{z_1^2 + 2x\alpha z_1 + \alpha^2}{2(1-x^2)})dz_1 \\
&= \frac{\sigma_w^2}{2\pi\sqrt{1-x^2}} (e^{-\frac{\alpha^2}{1+x}} - e^{-\frac{\alpha^2}{1-x}})
\end{aligned}
$$

we show a similar result for the second part and we conclude. $\qquad\square$

We proved that $f'' > 0$ for Hard-Tanh, all other conditions of proposition 5 (excluding the conditions on $\phi''$ since those were only used to prove $f'' > 0$) are verified, therefore the result of Proposition 4 holds for Hard-Tanh. we want to compare Tanh and Hard-Tanh when $\sigma_b$ is small since this is the important case. The proof of Proposition 4 gives us an idea on how to compare them, the ratio $\frac{\sigma_b^2}{q}$ controls the quality of approximation of f by the identity function, so a smaller ratio means a better approximation. Figure 2 shows that Tanh outperforms Hard-Tanh in this sense. This also means that for the same quality of approximation, we have bigger $q$ (bigger output variance) with Tanh compared to Hard-Tanh. This can particularly be due to the non-smoothness of Hard-Tanh, which slows down the dominated convergence in the proof of Proposition 4.

## C SUPPLEMENTARY EXPERIMENTAL RESULTS

### C.1 ELU ACTIVATION

We show numerically that the activation function ELU defined by $\phi(x) = (e^x - 1)1_{x<0} + x1_{x \geq 0}$ satisfies the conditions of Proposition 4. We have $\phi(x) = 0$, $\phi'(0^+) = 1$, $\phi'(0^-) = 1$ and $\left|\frac{\phi(x)}{x}\right| \leq 1$. Other conditions of Proposition 4 are shown numerically in graphs below.

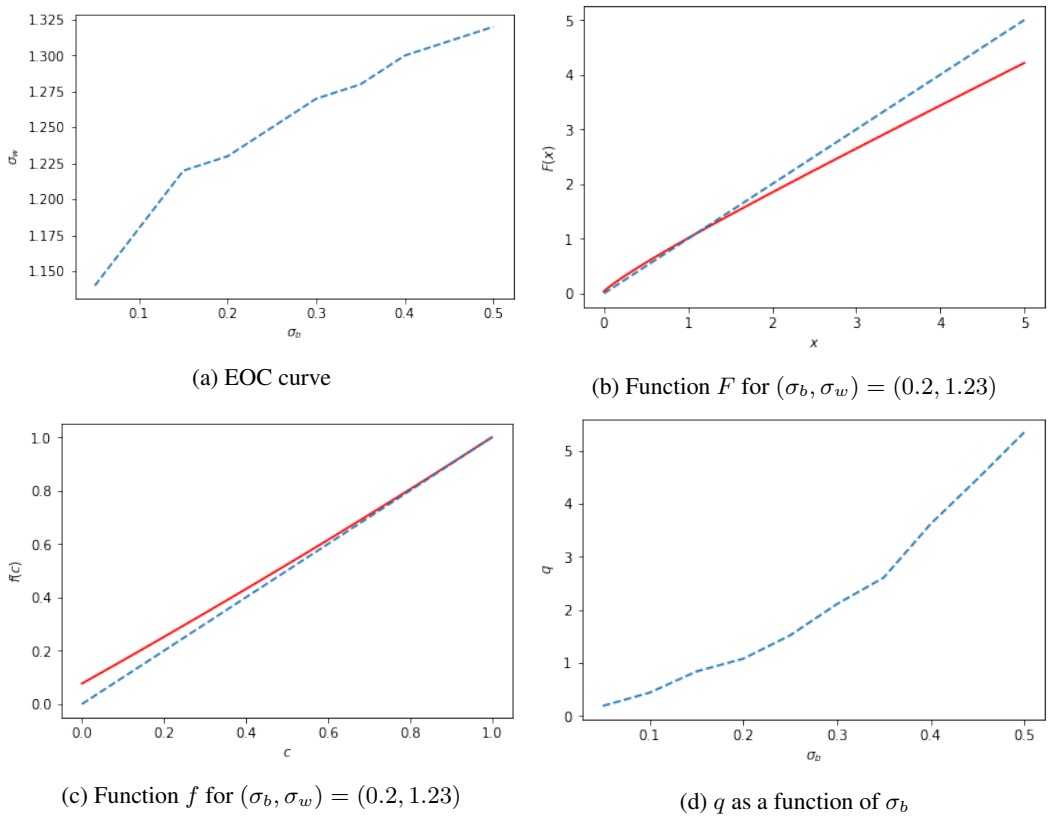

(a) EOC curve

(b) Function $F$ for $(\sigma_b, \sigma_w) = (0.2, 1.23)$

(c) Function $f$ for $(\sigma_b, \sigma_w) = (0.2, 1.23)$

(d) $q$ as a function of $\sigma_b$

Figure 12: Experimental results for ELU activation

Figure 12a shows the EOC curve (condition (ii) is satisfied). Figure 12b shows that is non-decreasing and Figure 12d illustrates the fact that $\lim_{\sigma_b \to 0} q = 0$. Finally, Figure 12c shows that function $f$ is convex. Although the figures of $F$ and $f$ are shown just for one value of $(\sigma_b, \sigma_w)$, the results are true for any value of $(\sigma_b, \sigma_w)$ on the EOC.

## C.2 WHAT HAPPENS WHEN THE DEPTH IS LARGER THAN THE WIDTH?

Table 2 presents a comparative analysis of the validation accuracy of ReLU and Swish when the depth is larger than the width, in which case the approximation by a Gaussian process is not accurate (notice that in the approximation of a neural network by a Gaussian process, we first let $N_l \to \infty$, then we consider the limit of large $L$). ReLU tends to outperforms Swish when the width is smaller than the depth and both are small, however, we still observe a clear advantage of Swish for deeper architectures.

Table 3: Validation accuracy for different values of (width, depth)

|       | (5,10)  | (10,20) | (30,40) | (40,50) |
|-------|---------|---------|---------|---------|
| ReLU  | **86.65** | **93.76** | 93.59   | 90.77   |
| Swish | 86.56   | 93.21   | **96.78** | **97.08** |

