# OpenReview forum: "On the Selection of Initialization and Activation Function for Deep Neural Networks"
_ICLR.cc/2019/Conference_

### Official Review · AnonReviewer3 · 2018-11-01
**Good results; providing some insights on the selection of activation function.**

**Rating:** 5
**Confidence:** 4

**Review:**

Good results; providing some insights on the selection of activation function.

This paper builds upon two previous works B.Poole etc. and S.S. Schoenholz etc. who initialized the study of random initialized neural network using a mean field approach (or central limit theorem.)
The two principal results of this paper are
1. Initializing the network critically on the edge of chaos.
2. Identifying some conditions on the activation functions which allow good "information flow" through the network.

The first result is not new in general (already appeared in Schoenholz etc. and many follow up mean field papers). However, the results about ReLU (initializing (weigh_variance, bias_variance)=(2, 0)) seems to be new. The author also shows that the correlations converge to 1 at a polynomial rate (proposition 3), which is interesting.

The second one is a novel part of this paper (proposition 5). If I understand correctly, the authors are trying to identify a class of activation functions (and suitable hyper-parameters) so that the network can propagate the sample-to-sample correlations (i.e. kernel) almost isometrically (please correct me if I am wrong). This is only possible 1) the activation functions are linear; OR 2) in the regime q->0, where the activation function has small curvature (i.e. almost linear). I think the results (and insights) are quite interesting. However, I don't think the authors provides enough theoretical or empirical evidence to support the claim that such activation functions can perform better.



cons:
1. I don't think the experimental results are convincing enough for the reasons below:
    1.1. All experiments are conducted over MNIST with testing accuracy around 96%.  The authors should consider using large datasets (at least Cifar10).
    1.2 The width (<=80) of the network is too small while the theory of the paper assumes the width approaches infinity. Width>=200 should be a reasonable choice. It should be possible to train a network with depth~200 and width ~200 and batch_size~64 in a single machine.
    1.3. Figure 6(b) seems unconvincing. ReLU network should be trainable with depth>=200; see figure 4 of the paper: "Resurrecting the sigmoid in deep learning through dynamical isometry: theory and practice"


2. The claim that swish is better than tanh because the latter suffers from vanishing of gradients is unconvincing. It has been shown in Schoenholz etc and many follow-up papers that ultra-deep tanh networks (>=1000 layers) can be trained with critical initialization.

3. Again, I don't think it is convincing to make the conclusion that swish is better than ReLU based on the empirical results on MNIST.

4. Using a constant learning rate (lr=0.001) for all depths (and all widths) is incorrect. I believe the gradients will explode as depth increases. Roughly, the learning rate should decay linearly with depth (and width) when the network is initialized critically.


In sum, the paper has some interesting theoretical results but the empirical results are not convincing.


Other comments:
1. The authors should explain the significance and motivation of proposition 4. In particular, explain why we need f(x)~x.
2. Consider replacing "Proposition 4" by  "Theorem", since it is the main result of the paper.

---

> ### Author Response · Authors · 2018-11-15
> **Re : Reviewer 3**
>
> The reviewer suggests that ‘However, I don't think the authors provides enough theoretical or empirical evidence to support the claim that such activation functions can perform better.’ In this paper, we only say that information propagation could partly explain why an activation function performs empirically better than another one. Of course, there are many other properties that make one activation better than another (e.g. vanishing gradient). As far as empirical evidence is concerned we refer the reader to Ramachandran et al. (2017). The present paper focuses on understanding/explaining/designing good initialization scheme for the algorithm, which is known to be crucial.
>
> 1) “All experiments are conducted over MNIST with testing accuracy around 96%. The authors should consider using large datasets (at least Cifar10).”: We are currently running more experiments right now. Note that an extensive set of simulations has already been performed in Ramachandran et al. (2017).
>
> “Figure 6(b) seems unconvincing. ReLU network should be trainable with depth>=200”. We do not claim that ReLU is not trainable in such scenarios. Figure 6-b only shows the first steps (40 epochs) of the training, because the information propagation analysis is only valid at the initial step, we are comparing the final accuracies here.
>
> 2) We only say that having an activation function that avoids the vanishing/exploding gradient problem is always better than having an activation function that has gradient strictly less than 1 (in absolute value) in deep neural networks. If a Tanh-network can be trained for 1000 layers, it does not mean those networks are not trainable with Swish or Relu.
>
> 4) We are currently running more experiments with different learning rates. We note that extensive experiments were already done in in Ramachandran et al. (2017) with different datasets and different learning rates.

---

### Official Review · AnonReviewer2 · 2018-11-02
**Interesting solid work but few concerns remain**

**Rating:** 4
**Confidence:** 4

**Review:**

Studying properties of random networks in the infinite width limit, this work suggests guidance for choosing initialization and activation function.

In my opinion, novel contribution comes for guidance for choosing activation functions and theoretical grounds for superior performance of ```'swish’ activation function.

I have two main concerns :

In terms of selection on initialization, the findings seem to be mostly discussed already in Schoenholz et al (2017) [1]. In their work, Edge of Chaos is critical line separating different phases and was already shown to have power-law decay rather than exponential decay. As far as I can tell, analysis on EOC on ReLU-like activations are different from Schoenholz et al (2017) [1]. Some of the results for ReLU are already appeared in the literature e.g. Lee et al (2018) [2].

Another main concern is in the author’s experimental setup. It is hard to draw conclusions when comparison experiments were done with a fixed learning rate.  As we know learning rate is one of the most critical hyperparameter for determining performance and optimal learning rate is often sensitive to architecture choice. Especially for different non-linearity and different depth/width optimal learning rate can change.

Pros:
 - Clearly written and easy to understand what authors are trying to say
 - Interesting theoretical support for activation function which recently got attention due to boosting performance in neural networks
 - Nice suggestion of choosing activation function for deep networks (Proposition 4)
      -- ELU/SELU/Softplus/Swish all satisfy this suggestion
Cons:
 - Novelty may be not strong enough as the standard analysis tool from [1] was mostly used
 - Experimental setup may suffer from some critical flaw

Few comments/questions:
- P3: Is M_{ReLU} = 2 correct, from ReLU EOC, shouldn’t it be ½?
- For all the works using activation functions satisfying Proposition 4 (ELU/SELU/Softplus/Swish), the initialization scheme close to EOC? Does this work’s analysis actually explain performance boost over ReLU for these activation functions?

[1] S.S. Schoenholz, J. Gilmer, S. Ganguli, and J. Sohl-Dickstein. Deep information propagation. 5th International Conference on Learning Representations, 2017.
[2] J. Lee, Y. Bahri, R. Novak, S.S. Schoenholz, J. Pennington, and J. Sohl-Dickstein. Deep neural networks as gaussian processes. 6th International Conference on Learning Representations, 2018.

---

> ### Author Response · Authors · 2018-11-15
> **Re : Reviewer 2**
>
> - Major concerns:
> * “In terms of selection on initialization, the findings seem to be mostly discussed already in Schoenholz et al (2017)”. We respectfully disagree with the reviewer. There are several original results in our manuscript. Among others, our main result -Proposition 4- provides sufficient conditions to ensure deep information propagation. This provides some theoretical grounding explaining the excellent empirical performance of Swish observed in several recent papers. We are not aware of any similar result in the literature.
>
> * We agree with the reviewer that more experiments could be done with different learning rates, which we are working on right now.  However, note that extensive experiments were already carried out in Ramachandran et al. (2017) with different datasets and different learning rates as indicated in our paper. We thus chose not to pursue more simulations in this paper and focused on theoretical properties enlightening the empirical findings of Ramachandran et al. (2017).
>
> - Other concerns:
> Few comments/questions:
> - P3: Is M_{ReLU} = 2 correct, from ReLU EOC, shouldn’t it be ½?: Yes indeed, that was a typo and it has been fixed, thank you for pointing this to us.
>
> - "For all the works using activation functions satisfying Proposition 4 (ELU/SELU/Softplus/Swish), the initialization scheme close to EOC? Does this work’s analysis actually explain performance boost over ReLU for these activation functions?": We believe that the fact that these activation functions satisfy Prop4 partially explain the performance boost, since it leads to a better initialization scheme which is known to let the information propagate deeper through the network.

---

### Official Review · AnonReviewer1 · 2018-11-04
**Some good ideas, many issues**

**Rating:** 3
**Confidence:** 5

**Review:**

The authors prove some theoretical results under the mean field regime and support their conclusions with a small number of experiments. Their central argument is that a correlation curve that leads to sub-exponential correlation convergence (edge of chaos) can still lead to rapid convergence if the rate is e.g. quadratic. They show that this is the case for ReLU and argue that we must ensure not only sub-exponential convergence, but also have a correlation curve that is close to the identity everywhere. They suggest activation functions that attain conditions as laid out in propositions 4/5 as an alternative.

The paper has many flaws:
- the value of the theoretical results is unclear
- the paper contains many statements that are either incorrect or overly sweeping
- the experimental setup and results are questionnable

Theoretical results:
**Proposition 1: pretty trivial, not much value in itself
**Proposition 2: Pretty obvious to the experienced reader, but nonetheless a valuable if narrow result.
**Proposition 3: Interesting if narrow result. Unfortunately, it is not clear what the ultimate takeaway is. Is quadratic correlation convergence "fast"? Is it "slow"? Are you implying that we should find activation function where at EOC convergence is slower than quadratic? Do those activation functions exist? It would be good to compare this result against similar results for other activation functions. For example, do swish / SeLU etc. have a convergence rate that is less than quadratic?
**Proposition 4: The conditions of proposition 4 are highly technical. It is not clear how one should go about verifying these conditions for an arbitrary activation function, let alone how one could generate new activation functions that satisfy these conditions. In fact, for an arbitrary nonlinearity, verifying the conditions of proposition 4 seems harder than verifying f(x) - x \approx 0 directly. Hence, proposition 4 has little to no value. Further, it is not even clear whether f(x) - x \approx 0 is actually desirable. For example, the activation function phi(x)=x achieves f(x) = x. But does that mean the identity is a good activation function for deep networks? Clearly not.
**Proposition 5: The conditions of prop 5 are somewhat simpler than those of prop 4, but since we cannot eliminate the complicated condition (ii) from prop 4, it doesn't help much.
**Proposition 6: True, but the fact that we have f(x) - x \approx 0 for swish when q is small is kind of obvious. When q is small, \phi_swish(x) \approx 0.5x, and so swish is approximately linear and so its correlation curve is approximately the identity. We don't need to take a detour via propposition 4 to realize this.

Presentation issues:
- While I understand the point figures 1, 2 and 4b are trying to make, I don't understand what those figures actually depict. They are insufficiently labeled. For example, what does each axis represent?
- You claim that for ReLU, EOC = {(0,\sqrt{2})}. But this is not true. By definition 2, EOC is a subset of D_\phi,var. But {(0,\sqrt{2})} is not in D_\phi,var, because it simply leaves all variances unchanged and does not cause them to converge to a single value. You acknowledge this by saying "For this class of activation functions, we see (Proposition 2) that the variance is unchanged (qal = qa1) on the EOC, so that q does not formally exist in the sense that the limit of qal depends on a. However,this does not impact the analysis of the correlations." Section 2 is full of complicated definitions and technical results. If you expect the reader to plow through them all, then you should really stick to those definitions from then on. Declaring that it's fine to ignore your own definitions at the beginning of the very next section is bad presentation. This problem becomes even worse in section 3.2, where it is not clear which definition is actually used for EOC in your main result (prop 4), making prop 4 even harder to parse than it already is.

Correctness issues:
- "In this chaotic regime, it has been observed in Schoenholz et al. (2017) that the correlations converge to some random value c < 1" Actually, the correlation converges deterministically, so c is not random.
- "This means that very close inputs (in terms of correlation) lead to very different outputs. Therefore, in the chaotic phase, the output function of the neural network is non-continuous everywhere." Actually, the function computed by a plain tanh network is continuous everywhere. I think you mean something like "the output can change drastically under small changes to the input". But this concept is not the same as discontinuity, which has an established formal definition.
- "In unreported experiments, we observed that numerical convergence towards 1 for l ≥ 50 on the EOC." Covergence of a sequence is a property of the limit of the sequence, and not of the 50th element. This statement makes no sense. Also if you give a subjective interpretation of those experimental results, you should present the actual results first.
- "Tanh-like activation functions provide better information flow in deep networks compared to ReLU-like functions." This statement is very vague and sweeping. Also, one could argue that the fact that ReLU is much more popular and tends to give better results than tanh in practice disproves the statement outright.
- "Tanh-like activation functions provide better information flow in deep networks compared to ReLU-like functions. However, these functions suffer from the vanishing gradient problem during back-propagation" At the edge of chaos, vanishing gradients are impossible! As Schoenholz showed, at the edge of chaos, \chi_1=1, but \chi_1 is also the rate of growth of the gradient. Pascanu et al (2013) discussed vanishing gradients in RNNs, which is a different story.
- "Other activation functions that have been shown to outperform empirically ReLU such as ELU (Clevert et al. (2016)), SELU (Klambauer et al. (2017)) and Softplus also satisfy the conditions of Proposition 4 (see Supplementary Material for ELU)." Firstly, SeLU does not satisfy proposition 4. f(x) \approx x requires \phi to be close to a linear function in the range where the pre-activations occur. Since SeLU has a kink at 0, it cannot be close to a linear function no matter how small the pre-activations are. Secondly, softplus also doesn't satisfy proposition 4, as \phi(0) = 0 does not hold. Thirdly, this statement is too sweeping. If ELU / SELU / Softplus "outperform" ReLU, why is ReLU still used in practice? At best, those nonlinearities have been shown to outperform in a few scenarios.
- "We proved in Section 3.2 that the Tanh activation guarantees better information propagation through the network when initialized on the EOC." Prop 4 only applies in the limit as \sigma_b converges to 0. So you can't claim that you showed tanh as "better information propagation" in general.
- "However, for deeper networks (L ≥ 40), Tanh is stuck at a very low test accuracy, this is due to the fact that a lot of parameters remain essentially unchanged because the gradient is very small." But in figure 6b the accuracy for tanh is decreasing rapidly, so therefore the parameters are not remaining "essentially unchanged", as this would also cause the accuracy to remain essentially unchanged. Also, if the parameter changes are too small ... why not increase the learning rate?
- "To obtain much richer priors, our results indicate that we need to select not only parameters (σb , σw ) on the EOC but also an activation function satisfying Proposition 4." Prop 4 only applies when \sigma_b is small, so you additionally need to make sure \sigma_b small.
- "In the ordered phase, we know that the output converges exponentially to a fixed value (same value for all Xi), thus a small change in w and b will not change significantly the value of the loss function, therefore the gradient is approximately zero and the gradient descent algorithm will be stuck around the initial value." But you are using Adam, not gradient descent! Adam explicitly corrects for this kind of gradient vanishing, so a small gradient can't be the reason for the lack of training success.

Experimental issues:
- "We use the Adam optimizer with learning rate lr = 0.001." You must tune the learning rate independently for each architecture for an ubiased comparison.
- In figure 6b, why does tanh start with a high accuracy and end up with a low accuracy? I've never seen a training curve like this ... This suggests something is wrong with your setup.
- You should run more experiments with a larger variety of activation functions.

Minor comments:
- "Therefore, it is easy to see that for any (σb , σw ) such that F is increasing and admits at least one fixed point,wehaveKφ,corr(σb,σw) ≥ qwhereqistheminimalfixedpoint;i.e. q := min{x : F(x) = x}." I believe this statement is true, but I also think it requires more justification.
- At the end of page 3, I think \epsilon_r should be \epsilon_q

There are some good ideas here, but they need to be developed/refined/polished much further before publication. The above (non-exhaustive) list of issues will hopefully be helpful for this.


### Addendum ###
After an in-depth discussion with the authors (see below), my opinion on the paper has not changed. All of my major criticisms remain: (1) There are far easier ways of achieving f(x) ~ x than propositions 4/5/7, i.e. we simply have to choose \phi(x) approximately linear. (2) The experiments are too narrow, and learning rates are badly chosen. (3) The authors do not discuss the fact that as f(x) gets too close to x, performance actually degrades as \phi(x) gets too close to a linear function. (Many other criticisms also remain.)

The one criticism that the authors disputed until the end of the discussion is criticism (1). Their argument seems to hinge on the fact that their paper provides a path to construct activation function that avoid "structural vanishing gradients", which they claim 'tanh' suffers from. While they acknowledge that tanh does not necessarily suffer from "regular" vanishing gradients (as shown by "Resurrecting the sigmoid in deep learning through dynamical isometry: theory and practice" and "Dynamical Isometry and a Mean Field Theory of CNNs: How to Train 10,000-Layer Vanilla Convolutional Neural Networks"), they claim it suffers from structural vanishing gradients. I do not believe that there is such a thing as structural vanishing gradients. However, even if such a concept did exist, it falls on the the authors to provide a clear definition / explanation, which they neither do in the paper nor the rebuttal.

---

> ### Author Response · Authors · 2018-11-15
> **Re : Reviewer 1 (part 1)**
>
> - Reviewer’s comments on theoretical results :
> ** “Proposition 1: pretty trivial, not much value in itself ” We agree that the proposition is trivial and this is stated explicitly after the Proposition. However, we think that after defining the Domains of Convergence, it is logical to give some sufficient conditions for (sigma_b, sigma_w) to be in those domains of convergence.
>
> ** “Proposition 2: Pretty obvious to the experienced reader, but nonetheless a valuable if narrow result”.  This Proposition gives explicitly the edge of chaos for an important set of activation functions. This result is indeed useful since it provides the parameters (sigma_b, sigma_w) that one should use for ReLU, Leaky-RELU etc.
>
> ** “Proposition 3: Interesting if narrow result. Unfortunately, it is not clear what the ultimate takeaway is. Is quadratic correlation convergence "fast"? Is it "slow"? Are you implying that we should find activation function where at EOC convergence is slower than quadratic?”  It is suggested that that ‘it is not clear what the ultimate takeaway is’. In the text just before Proposition 3, we explain that this proposition establishes that the convergence rate of the correlation of a ReLU network is 1/l^2 instead of the exponential rate exp( -b l). Hence compared to the rate outside the edge of chaos, this is a slow rate, which in practice means that the correlation (the information) propagates deeper inside the network and allows for use of many more layers.
>  This proposition indeed only deals with RELU, which makes it seem narrow, RELU activation functions are however widely used in practice. It is expected that similar phenomena hold under RELU-like activation functions. We did not pursue this here since we propose other activation functions which we believe are better behaved – as explained by our theory (prop 4 and 5) and the empirical evidence provided in Ramachandran et al (2017).
>
> ** Proposition 4 : The reviewer suggests that “The conditions of proposition 4 are highly technical. It is not clear how one should go about verifying these conditions for an arbitrary activation function”. We agree that the conditions can be difficult to verify theoretically, but in many cases, they can be verified numerically; see the Appendix for the ELU activation function. Proposition 5 provides additionally a simpler set of assumptions, which has been verified for important activation functions.
>
> ** “…for an arbitrary nonlinearity, verifying the conditions of Proposition 4 seems harder than verifying f(x) - x \approx 0 directly. Hence, Proposition 4 has little to no value…” We disagree with this comment. First, it is difficult to see how one could verify that  f(x) - x \approx 0 as sigma_b goes to zero without having any condition on the limit of q. Recall that f depends on q, and q may diverge, hence the use of condition (iii) in Prop4. Moreover, condition (iv) is necessary for uniform convergence of the correlation function f to the identity function. We could have stated a weaker condition, but it be  would more complicated to verify numerically.
>
> ** “it is not even clear whether f(x) - x \approx 0 is actually desirable. For example, the activation function phi(x)=x achieves f(x) = x. But does that mean the identity is a good activation function for deep networks? Clearly not.”: You are quite right the identity function, like any polynomial functions, is not a good activation function. We have explicitly mentioned in the introduction that for phi to be a `good’ activation function then it needs to be non-polynomial, phi should not suffer from the gradient vanishing problem and it should have a good information propagation, this last condition being the focus of our paper. Having  f(x) - x \approx 0 is a desirable property to obtain a good information propagation, since c^{l+1} = f(c^{l}). Therefore, having f close to the identity slows down the convergence of the correlation to 1, which means the information propagates deeper inside the network. The proof of Prop4 is informative in this respect.
> ** “Proposition 5: The conditions of prop 5 are somewhat simpler than those of prop 4, but since we cannot eliminate the complicated condition (ii) from prop 4, it doesn't help much.”: In Proposition 7 in the Appendix, we show how one replace condition (ii) of Prop4 by a simpler condition on phi. We will move this proposition to the main paper.

---

> > ### Comment · AnonReviewer1 · 2018-11-17
> > **Response**
> >
> > Dear Authors,
> >
> > Thank you for your comments.
> >
> > ** Proposition 1: To be clear, I wasn't trying to imply that it one shouldn't have "trivial" propositions in a paper. Proposition 1 is fine. My overall argument in this part of the review was that all theoretical results combined do not have enough impact, and to make that argument I looked at each individual result.
> >
> > ** Proposition 3: Yes, but we already knew that the convergence rate is not exponential. By the definition of the edge of chaos, the convergence is sub-exponential. I am asking: why is it important to know that for ReLU, the convergence is quadratic rather than some other sub-exponential rate.
> >
> > ** Proposition 4/5/7: I think there was a misunderstanding. I'm not saying it's easy to verify f(x) ~ x in the limit as \sigma_b converges to zero, i'm saying it's easy to verify f(x) ~ x for a fixed triplet (\phi, \sigma_w, \sigma_b). I.e. given some (\phi, \sigma_w, \sigma_b), I can verify whether f(x) ~ x holds true. And it's also easy to construct such a triplet. For example: pick any \phi that's close a linear function, initialize \sigma_b to zero and \sigma_w to some value for which a q* exists and isn't too large. We immediately get f(x) ~ x. If we also want to be on the edge of chaos, we can e.g. pick a \phi with \phi(x) = - \phi(-x), which ensures we are in the chaotic regime with \sigma_b=0. Then we can slowly increase \sigma_b from zero while decreasing \sigma_w to maintain q* until we hit the edge of chaos. All of this doesn't require the theory you present, doesn't require dealing with limits, doesn't require dealing with very small or very large q* etc.
> >
> > **My point with bringing up the identity activation function as a counterexample was to show that your implicit claim "f(x) ~ x makes for good activation functions" is not accurate. Sure you can say "well we can exclude the special case \phi(x) = x". But what about \phi(x) = x + 0.0000000000001 cos(x)? What about \phi(x) = x + 0.000000000000001 \sqrt(x)? Your theory is not robust.
> >
> > **Proposition 6: I disagree with your phi(x) = max(x+1,0) – 1 \approx x counterexample. When \sigma_w is just above 1, \sigma_b just above 0, we can be on the edge of chaos, have f(x) ~ x and q* small all at the same time. Also, my point is that if we want to verify f(x) ~ x for swish with small q*, we can do so directly, we don't even need to go via proposition 4 at all. And if we don't do that, the intuition that swish is close to a linear function is sufficient.
> >
> >
> > * Regarding figure 1, it might help to make it more explicit that d is the input dimension, that the output dimension is 1, and what the axes mean.
> >
> > * Nowhere in the statement " Therefore, in the chaotic phase, the output function of the neural network is non-continuous everywhere." does it mention infinite width / depth. It's not even clear what the definition of the limit of the output function as width / depth go to infinity would be.
> >
> > * The terms 'tanh-like' and 'ReLU-like' are both vague (what do they mean?) and sweeping (depending on how these terms are interpreted, they could include things that go far beyond your theory)
> >
> > * You comment regarding the fact that the edge of chaos is only achieved for tanh at the beginning of training makes no sense, because the edge of chaos is only defined in the randomly initialized state. A trained network does not follow the mean field model for any activation function. Also, "Dynamical Isometry and a Mean Field Theory of CNNs: How to Train 10,000-Layer Vanilla Convolutional Neural Networks" showed that one can indeed train arbitrarily deep nets with tanh if we initialize on the edge of chaos. This is not possible with ReLU. So with proper initialization and learning rate, tanh outperforms ReLU beyond a certain depth.
> >
> > * Adam is not equivalent to SGD, even at the first step, at least using the algorithm given in the original Adam paper. And even if it were, my criticism still holds for all other training steps.

---

> > > ### Comment · AnonReviewer1 · 2018-11-17
> > > **Additional Response**
> > >
> > >
> > > Finally, let me say that this paper could be very nice if it did the following:
> > > - Develop a theory for when (\phi, \sigma_w, \sigma_b) triplets have f(x) ~x, not in some strange limit, but just for specific triplets. Provide a description of the entire set of such triplets or at least a large space of such triplets. Make that description simple enough that it is insightful. My intuition is that you need something like a) phi(x) is close to some linear function lin(x) b) phi(x) - lin(x) is Lipschitz
> > > - develop a theory for how f(x) ~x is useful that makes specific predictions that can be tested in experiments. Discuss the limits of this theory (e.g. when \phi gets too close to a linear function, we suffer a loss of representational capacity)
> > > - provide an extensive empirical study covering a large range of (\phi, \sigma_w, \sigma_b) triplets that cover a range of sup(f(x) - x) values as well as a range of correlation convergence rates at the edge of chaos. Show how both properties affect performance. Tune learning rate extensively and independently for each setting. Ideally try several training algorithms.

---

> > > ### Author Response · Authors · 2018-11-17
> > > **Re : Reviewer 1**
> > >
> > > Thank you for your comments.
> > >
> > > *** Proposition 3 : this prop just gives the exact rate of convergence of the correlation. For experienced reader, this can be compared with the quadratic rate of convergence of the correlation for a residual neural network with ReLU activation.
> > >
> > > *** Proposition 4/5/7 : we agree that ‘given some (\phi, \sigma_w, \sigma_b), I can verify whether f(x) ~ x holds true ’. However, the suggested method of constructing such triplet has many issues. You suggest that ‘For example: pick any \phi that's close a linear function, initialize \sigma_b to zero and \sigma_w to some value for which a q* exists and isn't too large. We immediately get f(x) ~ x. If we also want to be on the edge of chaos, we can e.g. pick a \phi with \phi(x) = - \phi(-x), which ensures we are in the chaotic regime with \sigma_b=0. ’. First, it makes no sense to choose an activation close to a linear function, cause the neural network model will be equivalent to a simple linear regression.. so we assume you meant an activation function close to linear near 0. By initializing sigma_b to 0, 0 will be a fixed point of F, and q=0 for any sigma_w unless F is nearly concave near 0, which would imply (reader can check this) that the activation function has exploding gradients near 0. As an example, To have small q different from 0, you want F to be e.g. \F(x) = \sigma_b^2 + x^{\alpha} where 0<\alpha<1) .  However, this would lead to a problem of exploding gradient. Indeed, to satisfy this condition, we need to have (assuming \phi is differentiable a.e. and has a second derivative at least in the distribution sense) \sigma_w^2 \frac{1}{\sqrt{x}} \mathbb{E}[Z \phi'(\sqrt{x}Z)\phi(\sqrt{x}Z)] \approx \alpha x^{\alpha -1}, therefore, using Gaussian integration by parts, this yields \sigma_w^2 (\mathbb{E}[\phi'(\sqrt{x}Z)^2]+\mathbb{E}[\phi''(\sqrt{x}Z)\phi(\sqrt{x}Z)]) \approx \alpha x^{\alpha -1}. By letting x goes to 0 and knowing that \phi(0)=0, we have \lim_{y → 0} |\phi'(y)|=\infty (an example of such functions is \phi(y)=\sqrt{y}(1_{y>0} - 1_{y\leq 0})). Having \phi'(0) infinite is problematic as then the gradient explodes (for small values of y, \phi'(y) will be very large causing instabilities during back-propagation): this is why we enforce the condition \frac{\phi(x)}{x}|<K in Prop 4. It can be proved more generally for that taking F to be a concave function will lead to a problem of gradient exploding near 0.
> > > So your intuition of changing sigma_w until hitting the edge of chaos is wrong, cause you will always have q = 0 for any sigma_w.
> > >
> > >
> > > *** We understand your concern. By saying that we don’t consider activation functions that are trivial, we mean all activation functions that are close to polynomial activations. We will modify the wording.
> > >
> > > *** ‘You comment regarding the fact that the edge of chaos is only achieved for tanh at the beginning of training makes no sense, because the edge of chaos is only defined in the randomly initialized state. A trained network does not follow the mean field model for any activation function. ’ By beginning of the training we mean the first step, which is the initialization step, so we think we are saying the same thing.. could you explain why it makes no sense ?
> > > The paper "Dynamical Isometry and a Mean Field Theory of CNNs: How to Train 10,000-Layer Vanilla Convolutional Neural Networks" showed indeed that we can train CNN of 10000 layers with tanh activation by initializing on the EOC. However, note that here, it is not only the EOC that made it possible to train this very deep networks, it was a combination of a good choice of the learning rate, a good choice of the training algorithm, and finally a good initialization.
> > > ‘This is not possible with ReLU. So with proper initialization and learning rate, tanh outperforms ReLU beyond a certain depth. ‘ Is there a theoritical that proves this ? As far as we know, we are not sure if we can train such very deep networks with ReLU or not. Proving that it is not possible with ReLU needs more theoretical work.
> > >
> > > ** ‘Adam is not equivalent to SGD, even at the first step, at least using the algorithm given in the original Adam paper ’ Could you explain why ? In the original paper, as we said, the Adam optimizer initializes the first moment and second moments to m_1 = 0 and m_2=0, so the first step of the Adam optimizer is equivalent to an SGD step.
> > > ‘And even if it were, my criticism still holds for all other training steps’ Here we were addressing only your concern regarding the first step.

---

> > > > ### Comment · AnonReviewer1 · 2018-11-19
> > > > **Response**
> > > >
> > > > Thank you for your comments.
> > > >
> > > > (I will only respond to points raised for which I have new things to say. On other points, I will refer back to my earlier comments.)
> > > >
> > > > "First, it makes no sense to choose an activation close to a linear function, cause the neural network model will be equivalent to a simple linear regression.." Every activation function with f(x) ~ x is close to a linear function in the range where the pre-activations occur, i.e. something like [-3q*,3q*]. It's a matter of *how* close it is. If it is too close, then we get degraded performance as you describe. But if it is close but not *too* close, then we get the beneficial behavior you describe in your paper. My criticism is precisely that you don't make this distinction clear in the paper.
> > > >
> > > > " However, this would lead to a problem of exploding gradient. " Yes, \sigma_b=0 and q* > 0 implies exploding gradients. That's why I said this construction puts us in the chaotic regime. But that's not a problem if the explosion is not too fast relative to the depth of the net. If you have f(x) ~ x, then the explosion rate will be close to 1, so likely not an issue. If you absolutely want stable gradients, then, as I suggested, you can increase \sigma_b and decrease \sigma_w until you hit the edge of chaos.
> > > >
> > > > "So your intuition of changing sigma_w until hitting the edge of chaos is wrong, cause you will always have q = 0 for any sigma_w." I never said change only sigma_w, I said change sigma_w AND sigma_b. Let me be more specific. Take some triplet (\phi,\sigma_w,\sigma_b), assume \sigma_b=0, assume q* = \sigma_w^2 F(q*) > 0 and assume that at q*, we are in the chaotic regime / have exploding gradients. Then for all triplets (\phi,\sigma'_w,\sigma'_b) with \sigma_w^2 F(q*) = \sigma'_w^2 F(q*) + \sigma'_b^2, we have q* = \sigma'_w^2 F(q*) + \sigma'_b^2 for the same q*. Thus out of all triplets (\phi,\sigma'_w,\sigma'_b) we simply pick the one that is at the edge of chaos. This must exist because the set of explosion rates \chi achievable with this construction varies between the explosion rate for (\phi,\sigma_w,\sigma_b), which we assumed was greater 1, and the explosion rate for (\phi,0,\sigma_w \sqrt{F(q*)}), which is zero. Hence some intermediate explosion rate is exactly one.
> > > >
> > > > By the way, why are polynomial activation functions bad? I've recently trained some nets with the x^2 activation function and that worked ok.
> > > >
> > > > "By beginning of the training we mean the first step, which is the initialization step, so we think we are saying the same thing.. could you explain why it makes no sense ?" OK, I think I misunderstood your original statement. I think we are saying the same thing. Apologies.
> > > >
> > > > "However, note that here, it is not only the EOC that made it possible to train this very deep networks, it was a combination of a good choice of the learning rate, a good choice of the training algorithm, and finally a good initialization." So? Every network requires a good choice of learning rate. AFAIK, they used a standard training algorithm. Regarding choosing a good initialization ... isn't that one of the messages of your paper?
> > > >
> > > > "Is there a theoritical that proves this ? As far as we know, we are not sure if we can train such very deep networks with ReLU or not. Proving that it is not possible with ReLU needs more theoretical work." Isn't one of your core message that ReLU doesn't work for networks beyond a certain depth ... ?? it's precisely because at all fixed points of F, f(x) is far away from x.

---

> > > > > ### Author Response · Authors · 2018-11-19
> > > > > **Re : Reviewer 1**
> > > > >
> > > > > Thank you for your comments.
> > > > >
> > > > > ** “Yes, \sigma_b=0 and q* > 0 implies exploding gradients. That's why I said this construction puts us in the chaotic regime.” Note that there are two different notions of exploding gradients: exploding gradient linked to the chaotic phase, and the exploding gradient linked to a structural property of phi. It is not clear which one you are mentioning. To make our answer to your earlier comments more precise in this regard:
> > > > > In our previous answer, we showed that taking sigma_b = 0 would leave with two options : if the function F is convex near 0 (which is the case for tanh, swish etc), either you choose sigma_w two small such that F’(0)<1 and in this case q*=0 (since 0 is a an attractive fixed point) or you choose sigma_w such that F’(0)>1 and in this case q^l will most likely diverge. To avoid this problem, you want the function F to be nearly concave near 0 in order to have another stable fixed point (other than zero), but as we showed in our previous response, this would lead to am structural problem in your activation function since it will cause an exploding gradient phenomenon for pre-activations close to 0 (this is different from the exploding gradient linked to the chaotic phase).
> > > > > However, in general, your way of finding sigma_b and sigma_w that hit the EOC is exactly what we have used in practice.
> > > > >
> > > > >
> > > > > ** “By the way, why are polynomial activation functions bad? I've recently trained some nets with the x^2 activation function and that worked ok.” Because in this case the output of the network is polynomial function of order 2^L where L is the number of layers. So you’re basically doing polynomial regression of order 2^L. Why not doing it directly ? There is a well developed theory for this purpose. Moreover, by doing such regression with a neural network, you are restricting the expressive power of this regression because the output weights are functions of the weights inside the network. Of course, you will have less weights compared to the standard polynomial regression, but this is only because you are assuming such restrictions on the weights, and it is not intuitive why taking such functions would be better than doing e.g. a Lasso regularization to reduce the dimension and select only significant variables.
> > > > >
> > > > > ** “So? Every network requires a good choice of learning rate. AFAIK, they used a standard training algorithm. Regarding choosing a good initialization ... isn't that one of the messages of your paper?” Yes, choosing a good initialization is crucial for a good training. But, we are not saying it is the only thing that influence the training. In majority of cases, the learning rate and the ‘structural’ exploding gradient problem have more influence than the initialization. So we agree with you and are essentially saying the same thing
> > > > >
> > > > >
> > > > > ** “Isn't one of your core message that ReLU doesn't work for networks beyond a certain depth ... ?? it's precisely because at all fixed points of F, f(x) is far away from x.” Yes it is, but again we are just saying that ReLU suffers ‘more seriously’ from this problem of Information Propagation compared to Swich, Tanh etc .. However, it is not clear whether this problem would make it IMPOSSIBLE for a very deep ReLU network to be trained, in the same way that the gradient vanishing problem of Tanh (we are not talking about the initialization step, but during the training) didn’t make it impossible to train a very deep Tanh network.

---

> > > > > > ### Comment · AnonReviewer1 · 2018-11-19
> > > > > > **Response**
> > > > > >
> > > > > > I don't understand your argument regarding "structural" exploding gradient. Why not have F be concave and differentiable at 0? This is what happens for tanh. Then we don't need \phi(0) = \infty.
> > > > > >
> > > > > > I'm not sure about your argument regarding polynomial regression, though you may be right. If you have no citations for it, I wouldn't include it in the paper without further evidence. "Of course, you will have less weights compared to the standard polynomial regression" ... this kind of weight sharing is generally regarded as one of the positive features of deep nets.
> > > > > >
> > > > > > When I say "ReLU doesn't work for networks beyond a certain depth" I mean in the context of plain nets with standard training algos and no tricks, as studied in your and other mean field papers. So I think we're in agreement.

---

> > > > > > > ### Author Response · Authors · 2018-11-19
> > > > > > > **Re : Reviewer 1**
> > > > > > >
> > > > > > > Thank you for your comments.
> > > > > > >
> > > > > > >
> > > > > > > ** ' I don't understand your argument regarding "structural" exploding gradient. Why not have F be concave and differentiable at 0? This is what happens for tanh. Then we don't need \phi(0) = \infty.' In our previous response about this issue, our last comment was ' It can be proved more generally for that taking F to be a concave function will lead to a problem of gradient exploding near 0'. Actually, we forgot to mention a second case where this could lead to a 'problem of gradient vanishing'. Apologies for that. Now we explain the full problem in the two cases :
> > > > > > > -- First case : F'(0+) = \infty will lead to a problem of gradient exploding near 0 as previously discussed
> > > > > > > -- Second case : F'(0) is finite. There is too many cases to discuss here, but let's take the case of an activation such that phi''(x) small for all x (tanh for example).  Using the Gaussian integration by parts, we have F'(x) = \sigma_w^2 (\mathbb{E}[\phi'(\sqrt{x}Z)^2]+\mathbb{E}[\phi''(\sqrt{x}Z)\phi(\sqrt{x}Z)]). The second term is usually negligible (since phi'' is very small, this  is the case for tanh for example), F'(x) is essentially given by the first term. Having F' decreasing (F concave) would imply that phi' is decreasing and is 'more concentrated' near zero. Since the maximum at zero should does not usually exceeds 1 (otherwise, we will have an exploding gradient near 0) This would lead to a problem of vanishing gradient as we have said. For example, Tanh' has a maximum in zero, but the concavity of F forces the gradient of tanh to decreasing. The 'more' concave F is, the more decreasing ph' should be.
> > > > > > > So, we agree with your approach if you choose a tanh-like activation. However, as we discuss in our paper, we avoid activations that have this gradient vanishing problem (we believe this problem makes the training very slow if not impossible) .Moreove, this will not work with activations that have gradients approx to 1 at least on an subset of R with infinite measure (Swish, Elu etc)
> > > > > > >
> > > > > > >
> > > > > > > ** " this kind of weight sharing is generally regarded as one of the positive features of deep nets." We agree that for non polynomial activation this weight sharing is one of the positive features for deep nets. However, with polynomial activation, it is not clear why this would be better than a lasso regularization.. at least the second method would gives us Directly an idea about the features that are contributing the most to the output..

---

> > > > > > > > ### Comment · AnonReviewer1 · 2018-11-21
> > > > > > > > **Response**
> > > > > > > >
> > > > > > > > Why do you we care about the value of \phi' near 0? As long as \chi ~ 1, we are good. Exploding / vanishing gradient is determined by the size of \phi' across the spectrum of pre-activations, not just near 0. If q* is not small, then \chi and \phi'(0) are not the same. Also, the paper I cited did train very deep tanh networks, so it can be done successfully.

---

> > > > > > > > > ### Author Response · Authors · 2018-11-21
> > > > > > > > > **Re : Reviewer 1**
> > > > > > > > >
> > > > > > > > >  I think there is a misunderstanding. We don’t understand what your point is. To sum up the issue :
> > > > > > > > >
> > > > > > > > > (1) We want to avoid activation functions that suffer from the vanishing gradient problem (tanh for instance) which is a problem that is independent from the initialization. Avoiding this vanishing gradient problem is one of the main reasons why ReLU has a big success. This ‘structural’ gradient vanishing problem slows the training and sometimes makes it impossible. This problem was extensively studied, we cite for example :
> > > > > > > > > -  Krizhevsky et al. “ImageNet Classification with Deep Convolutional Neural Networks ”
> > > > > > > > > - Glorot et al. "Deep Sparse Rectifier Neural Networks"
> > > > > > > > > We also recommend checking this Notebook 'https://cs224d.stanford.edu/notebooks/vanishing_grad_example.html' to understand more this issue.
> > > > > > > > > In summary, tanh-like activations are not suited for deep neural networks because of this gradient vanishing problem during back-propagation.
> > > > > > > > >
> > > > > > > > > (2) Our point is to find a combination of good initialization and activation function (that does not suffer form the issue above). Swish is a good candidate : first because it doesn’t suffer from the gradient vanishing mentioned above (like ReLU), and second, because it allows better information flow (compared to ReLU) when initialized at the edge of chaos with small sigma_b.
> > > > > > > > >
> > > > > > > > > Your proposition is to choose an activation phi such that F is concave so we that we can have sigma_b=0 and q*>0. But, because of what we said in our previous comment, we believe that any activation function phi that satisfies ‘F is concave’ will suffer either from an exploding gradient near 0 or a ‘structural’ vanishing gradient (problem mentioned in (1)). As we said, the two cases :
> > > > > > > > >
> > > > > > > > >
> > > > > > > > > - First case : F'(0+) = \infty will lead to a problem of gradient exploding near 0. To see this, using Gaussian integration by parts, we have \sigma_w^2 (\mathbb{E}[\phi'(\sqrt{x}Z)^2]+\mathbb{E}[\phi''(\sqrt{x}Z)\phi(\sqrt{x}Z)]) = F’(x). By letting x goes to 0 and knowing that \phi(0)=0, we have \lim_{y → 0} |\phi'(y)|=\infty (an example of such functions is \phi(y)=\sqrt{y}(1_{y>0} - 1_{y\leq 0})). Having \phi'(0) infinite is problematic as then the gradient explodes (for small values of y, \phi'(y) will be very large causing instabilities during back-propagation)
> > > > > > > > >
> > > > > > > > >
> > > > > > > > > - Second case : F'(0+) is finite. There is too many cases to discuss here, but let's take the case of an activation such that phi''(x) small for all x (tanh for example). Using the Gaussian integration by parts, we have F'(x) = \sigma_w^2 (\mathbb{E}[\phi'(\sqrt{x}Z)^2]+\mathbb{E}[\phi''(\sqrt{x}Z)\phi(\sqrt{x}Z)]). The second term is usually negligible (since phi'' is very small, this is the case for tanh for example), F'(x) is essentially given by the first term. Having F' decreasing (F concave) would imply that phi' is decreasing and is 'more concentrated' near zero. This would lead to a problem of vanishing gradient as we have said.
> > > > > > > > >
> > > > > > > > > Your method of choosing the triplet (sigma_b, sigma_w, phi) will be useful only if phi suffers from problem (1), which we trying to avoid and we made that clear in the introduction of the paper.
> > > > > > > > >
> > > > > > > > > To sum up, we don’t understand your concern with proposition 4, as we have said, it is very easy to verify those conditions numerically, we gave an example for ELU and Swish, and we provided numerical evidence that Swish is better than ReLU.

---

> ### Author Response · Authors · 2018-11-15
> **Re : Reviewer 1 (part 2)**
>
> ** “Proposition 6: True, but the fact that we have f(x) - x \approx 0 for swish when q is small is kind of obvious. When q is small, \phi_swish(x) \approx 0.5x, and so Swish is approximately linear and so its correlation curve is approximately the identity. We don't need to take a detour via proposition 4 to realize this.”: We believe that \phi(x) being approximately linear near zero is not good enough for the results of Prop4 to hold and might lead to wrong intuition. An important condition is that we can find a regime where q is small. The condition ‘q is small’ is not necessarily guaranteed. The condition on the limit of q as sigma_b goes to zero is precisely condition (iii) of Prop4. Having only the property that phi is linear near 0 is not sufficient to ensure that the result of Prop4 holds. For example, the function phi(x) = max(x+1,0) – 1 \approx x near 0. However, one can check that q does not converge to 0 in this scenario so that the result of prop 4 do not hold.
>
> - Reviewer’s comments on presentation issues:
> * We apologize if fig 1 was not so easy to understand. It displays a draw of an output of a RELU (resp Tanh) network, as such it is a function from [0,1]^2 to \R (in both cases). We mentioned in the text of page 3 that Figure 1 illustrates this behaviour for d = 2 for ReLU and Tanh using a network of depth L = 10 with Nl = 100 neurons per layer. The axes are x and y (belonging to [0,1]^2) in dimension d=2. Other figures have the same axes.
>
> * We agree that definition of the EOC is not relevant for ReLU. We could make the definition of the EOC more general to include ReLU-like functions. We thought it was an unnecessary complication and this ‘modification’ was introduced only in the section of ReLU-like activations. We agree that this change of definition is actually confusing and we will address it.
>
>
> - Reviewer’s comments on correctness issues:
> * By ‘some random value c < 1’: we meant a value of c in the interval (0,1), we will modify the wording.
>
> * “Actually, the function computed by a plain Tanh network is continuous everywhere. I think you mean something like "the output can change drastically under small changes to the input". We agree with the reviewer that the output is continuous when the network has finite width and depth. However, in the limit of infinite width and depth, it can be proven that the output function is discontinuous everywhere.
>
> * By ‘we observed that numerical convergence towards 1 for l ≥ 50 on the EOC.’ we meant that output start to be visually constant around l = 50.
> * We disagree that the statement "Tanh-like activation functions provide better information flow in deep networks compared to ReLU-like functions" is very vague and sweeping. We are not claiming that “Tanh is better than ReLU”. We are just saying that Tanh verifies conditions of Prop4, therefore it provides better information propagation compared to ReLU which does not enjoy this property.
>
> * “At the edge of chaos, vanishing gradients are impossible! As Schoenholz showed, at the edge of chaos, \chi_1=1, but \chi_1 is also the rate of growth of the gradient.” The result stated from Schoenholz et al. is correct but the vanishing gradient problem is not avoided by initializing the network on the on the edge of chaos. Such an initialization only addresses the vanishing gradient problem at the first step of the learning algorithm. In the subsequent steps of gradient descent, this problem will occur again as the weights are no longer ‘distributed’ on the edge of chaos.
> * ”In the ordered phase, we know that the output converges exponentially to a fixed value (same value for all Xi), thus a small change in w and b will not change significantly the value of the loss function, therefore the gradient is approximately zero and the gradient descent algorithm will be stuck around the initial value." But you are using Adam, not gradient descent! Adam explicitly corrects for this kind of gradient vanishing, so a small gradient can't be the reason for the lack of training success.’   The Adam optimizer initializes the first moment and second moments to m_1 = 0 and m_2=0, so the first step of the Adam optimizer is equivalent to the first step of usual gradient descent.
>
> - Setup issues:
> * Thanks for your comment about figure 6-b. Indeed, after running a second experiment, the drop in the curve of Tanh seems to be due to a learning rate issue, however, the conclusion ‘ReLU is better than Tanh’ in this case is still numerically true.
> * We are currently running experiments with different learning rates. Note that many experiments were already done in Ramachandran et al. (2017) with different datasets and different learning rates as indicated in the paper.

---

### Meta-Review · Area_Chair1 · 2018-12-17
**Lack of clarity makes it difficult to discern a clear contribution over recent literature**

**Confidence:** 5
**Recommendation:** Reject

**Metareview:**

The paper attempts to extend the recent analysis of random deep networks to alternative activation functions.  Unfortunately, none of the reviewers recommended the paper be accepted.  The current presentation suffers from a lack of clarity and a sufficiently convincing supporting argument/evidence to satisfy the reviewers.  The contribution is perceived as too incremental in light of previous work.